# Differential and shared genetic effects on kidney function between diabetic and non-diabetic individuals

Reduced glomerular filtration rate (GFR) can progress to kidney failure. Risk factors include genetics and diabetes mellitus (DM), but little is known about their interaction. We conducted genome-wide association meta-analyses for estimated GFR based on serum creatinine (eGFR), separately for individuals with or without DM ($n_{DM} = 178{,}691$, $n_{noDM} = 1{,}296{,}113$). Our genome-wide searches identified (i) seven eGFR loci with significant DM/noDM-difference, (ii) four additional novel loci with suggestive difference and (iii) 28 further novel loci (including *CUBN*) by allowing for potential difference. GWAS on eGFR among DM individuals identified 2 known and 27 potentially responsible loci for diabetic kidney disease. Gene prioritization highlighted 18 genes that may inform reno-protective drug development. We highlight the existence of DM-only and noDM-only effects, which can inform about the target group, if respective genes are advanced as drug targets. Largely shared effects suggest that most drug interventions to alter eGFR should be effective in DM and noDM.

Impaired kidney function can progress to kidney failure requiring dialysis or transplantation, which implicates high early mortality[1] and public health burden[2]. The most common kidney function measure is the glomerular filtration rate (GFR). Chronic kidney disease (CKD) can be defined as GFR < 60 ml/min per 1.73 m$^2$, and has a prevalence of ~10% in the population[3]. In large-scale epidemiological studies, GFR can be estimated from serum creatinine (eGFR) using the CKD-Epi formula[4].

Diabetes mellitus (DM) is a major risk factor for impaired kidney function[5] and is present in 30% of individuals starting dialysis in Europe[6]. Most current pharmacotherapy in DM has substantial side effects[7–10]. Recent work showed that SGLT2 inhibitors and GLP1R are reno-protective for individuals with DM[11,12] and early evidence indicates that this might be also true for individuals without DM (noDM)[13]. The biological processes related to lower kidney function might differ between individuals with DM and individuals without DM. Understanding the mechanisms of kidney function variability within and between these groups is pivotal for understanding pathogenesis. Genome-wide association studies (GWAS) have pinpointed hundreds of loci for eGFR[14–16]. These loci help understand kidney function variability in the overall population. However, commonalities and differences in genetic kidney function effects between DM and noDM individuals are not well understood.

Genes underneath GWAS loci have been shown to double the success rate in drug development pipelines[17,18]. This renders each gene in eGFR loci a drug target to alter kidney function, particularly when the gene maps to association-driving variants that are relevant for the protein or for the gene's expression in kidney tissue[19]. This opens up a clinically important question in the context of differential or shared genetic kidney function effects between DM and noDM individuals: when a locus identified for eGFR in the overall population turns out to have no effect in DM individuals, the underlying causal gene might not be informative for reno-protection in DM individuals. This is not an unlikely scenario, since only 10% of individuals in previous population-based GWAS on eGFR overall had DM[16,20].

Previous DM-/noDM-stratified searches for genetic eGFR-effects have been limited[21] and without identification of a significant DM/noDM-difference. It is unclear whether this was due to limited power or lack of such differential effects. GWAS for eGFR-based kidney function outcomes in DM, used as one approach to identify the genetics of Diabetic Kidney Disease (DKD), identified four loci in ~40,000 type 1 or type 2 DM patients[22] and four other loci in ~19,500 type 1 DM patients[23]. When genetic effects on eGFR in DM versus noDM have little overlap, such focused searches will remain the best option. For shared genetics, GWAS on eGFR overall will have much better power.

The quest for understanding commonalities and differences of genetic eGFR-effects between DM and noDM has two aspects: a search for eGFR-associated loci with DM/noDM-difference will help understand whether such differential effects exist and a search for SNP-association on eGFR allowing for DM/noDM-difference may detect novel eGFR loci. Statistically, DM/noDM-difference in eGFR association and genetics-by-DM-status interaction on eGFR association are equivalent. There are various approaches to do so[24]. These approaches have been applied successfully for other phenotypes[25–29], but not yet for GWAS interaction analyses for kidney function.

We thus set out to search for eGFR loci with DM/noDM-difference and for novel eGFR loci allowing for difference. For this, we gathered GWAS data on eGFR separately for 178,691 DM and 1,296,113 noDM individuals. We prioritized genes underneath identified loci using in-silico functional evidence. We also evaluated the impact of DM-/noDM-specific weights on the genetic risk score (GRS) for eGFR in data independent from the variant- and weight-identifying step[30].

## Results

**Overview of the GWAS meta-analyses.** We first analyzed GWAS data on 7,046,926 single nucleotide polymorphisms (SNPs) and their association with logarithm-transformed eGFR separately for 109,993 DM (type 2 or type 1) and 1,070,999 noDM individuals (stage 1, including 72 studies from CKDGen Consortium[31] and UK Biobank, UKB[32]; mostly European-ancestry; Supplementary Data 1, Methods, Fig. 1a). These DM-/noDM-stratified GWAS summary statistics allowed us to apply the difference test, the joint test and the stratified tests (Methods). Based on these tests, we searched for eGFR-associated loci with DM/noDM-difference and for novel eGFR-associated loci allowing for DM/noDM-difference.

We sought replication of identified loci in independent DM-/noDM-stratified data (stage 2, $n_{DM} = 68,698$, $n_{noDM} = 225,114$; Million Veterans Program, MVP[33], Michigan Genomics Initiative, MGI[34], Trøndelag health study, HUNT[35]; all European-ancestry; Supplementary Data 1, Methods). While the discovery+replication design augments confidence in identified loci, this comes at the cost of lower power[36]. We also conducted GWAS searches in a combined stage design (stage $1 + 2$: $n_{DM} = 178,691$, $n_{noDM} = 1,296,113$), to yield significant loci exploiting the full sample size, yet without independent replication.

The two stages of data were also used to separate the variant identification and weight quantification for the GRS (stage 1 data) from the GRS association analyses (stage 2 data). The analysis workflow is shown in Fig. 1.

**Seven eGFR loci identified with DM-/noDM-differential effects.** To search for eGFR loci with DM/noDM-difference, we applied two approaches[24,37] (Fig. 1b): (i) a genome-wide difference test ($P_{Diff} < 5 \times 10^{-8}$, difference test approach), and (ii) a search for genome-wide significant association with overall eGFR followed by a difference test in the same data ($P_{Overall} < 5 \times 10^{-8}$ and $P_{Diff} < 0.05/k$, k = number of followed SNPs, overall+difference test approach, Methods).

In the discovery search (stage 1, $n_{DM} = 109,993$; $n_{noDM} = 1,070,999$), we identified four eGFR loci with significant DM/noDM-difference (Supplementary Fig. 1): (i) two by the difference test approach (rs77924615 near UMOD-PDILT, rs12233328 near PDE9A; $P_{Diff} < 5 \times 10^{-8}$, Supplementary Data 2), and (ii) two further loci by the overall+difference test approach (near TPPP and MED1-NEUROD2; $P_{Overall} < 5 \times 10^{-8}$ and $P_{Diff} < 0.05/610 = 8.2 \times 10^{-5}$; corrected for 610 followed variants[16], Supplementary Data 2). Details of the overall+difference test results are provided in Supplementary Note 1 and Supplementary Data 3, conditional difference analyses on known $P_{Overall} < 5 \times 10^{-8}$ variants in UMOD-PDILT in Supplementary Fig. 2. In stage 2 data ($n_{DM} = 68,698$, $n_{noDM} = 225,114$), three of the four loci replicated (near UMOD-PDILT, MED1-NEUROD2, and TPPP; one-sided $P_{Diff} < 0.05/4 = 0.0125$; Table 1, Supplementary Data 2), while the PDE9A locus variant did not ($P_{Diff} = 0.30$).

In the combined stage design (stage $1 + 2$: $n_{DM} = 178,691$; $n_{noDM} = 1,296,113$), we identified seven eGFR loci with significant DM/noDM-difference (Table 1, Fig. 2a, Supplementary Fig. 3): (i) six loci by the difference test approach ($P_{Diff} < 5 \times 10^{-8}$, Supplementary Data 2) including the three loci already found by the discovery+replication approach now all with $P_{Diff} < 5 \times 10^{-8}$ (near UMOD-PDILT, TPPP, and MED1-NEUROD2;), and three additional loci (near CSRNP, DCDC5, and NRIP1), (ii) one locus

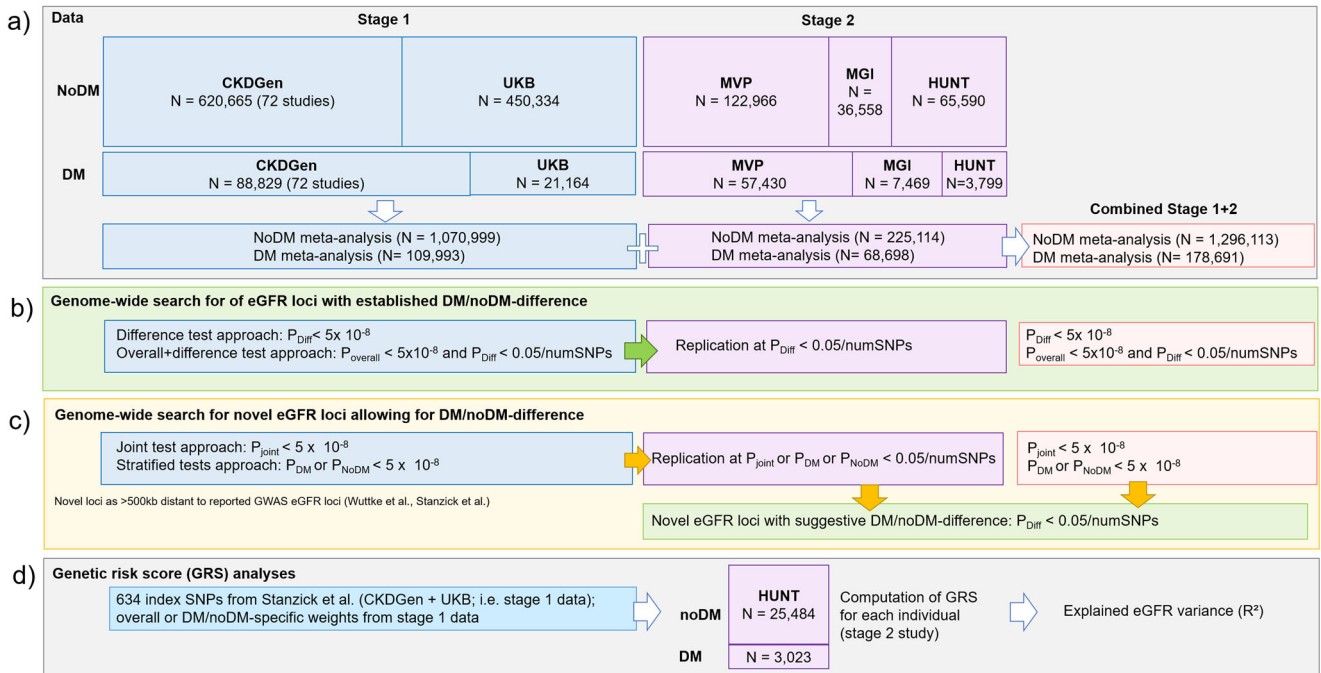

**Fig. 1 Data and analysis workflow. a** Overview on datasets and meta-analyses. **b** Approaches to identify DM/noDM-differences. **c** Approaches to identify novel eGFR loci and suggestive DM/noDM-differences. **d** Genetic risk score (GRS) analyses in HUNT.

from the overall+difference test approach (near *SLC22A2*; $P_{Overall} < 5 \times 10^{-8}$ and $P_{Diff} < 0.05/610 = 8.2 \times 10^{-5}$; 610 followed variants as described above, Supplementary Data 2).

Approximate conditional analyses using the difference test (combined stage 1 + 2, European only, Methods) did not identify any further independent variants within the 7 identified difference loci (regional difference plots in Supplementary Fig. 4). Interestingly, there was also no further independent variant with DM/noDM-difference in the *UMOD-PDILT* locus besides rs77924715, despite the multiple independent variants known for eGFR association overall[16,20] (Supplementary Fig. 4a, b). Sensitivity analyses confirmed that DM/noDM-difference was also observed without log-transformation of eGFR and also after adjusting for interaction with age or other eGFR risk factors (Supplementary Note 2, Supplementary Data 4, Supplementary Table 1).

With respect to DM-/noDM specificity and magnitude of effects, we found the following (Fig. 2b): (i) six of the seven variants identified with differential eGFR-effects showed more pronounced effects in DM versus noDM (near *UMOD-PDILT, TPPP, DCDC5, NRIP1,* and *SLC22A2*) or a DM-only effect on eGFR (near *CSRNP1*) and (ii) one variant showed a noDM-only effect (near *MED1-NEUROD2*). These patterns were consistent in stage 1 and stage 2 separately (Supplementary Fig. 5). Of note, with the combined stage 1 + 2 sample size, the difference detectable for a common variant of 30% minor allele frequency at 80% power was 0.0050 or 0.0037 log(ml/min/1.73 m²) for a DM-only or noDM-only effect, respectively (Supplementary Note 3, Supplementary Fig. 6).

In summary, we established seven eGFR loci with DM/noDM-difference (near *UMOD-PDILT, TPPP, MED1-NEUROD2, CSRNP1, DCDC5, NRIP1,* and *SLC22A2*) including one with DM-only (*CSRNP1*) and one with noDM-only effect (*MED1-NEUROD2*). While all seven difference loci were known loci for eGFR[16], these are the first eGFR loci reported with significant DM/noDM-difference from a genome-wide search for the difference to the best of our knowledge.

**Allowing for difference identifies 32 novel eGFR-associated loci including *CUBN*.** To search for novel eGFR-associated loci allowing for the difference between DM and noDM, we applied two approaches (Fig. 1c, Methods): (i) a genome-wide joint test[38], which considers the main SNP effect on eGFR and the SNP-by-DM-status interaction effect jointly (joint-test approach) and (ii) GWAS separately in DM or noDM (stratified tests approach).

In the discovery (stage 1; $n_{DM} = 109,993$, $n_{noDM} = 1,070,999$), we identified 25 novel eGFR loci after excluding previously identified GWAS eGFR loci[16,20] ($P_{Joint}$ or $P_{noDM} < 5 \times 10^{-8}$, none by $P_{DM} < 5 \times 10^{-8}$, Supplementary Data 5, Supplementary Fig. 7a). In stage 2 data ($n_{DM} = 68,698$, $n_{noDM} = 225,114$;), three of the 25 loci replicated ($P_{Joint}$ or $P_{noDM} < 0.05/25$; near *ATP12A, SERTAD2,* and *ABCC2*, Table 2).

In the combined stage design (stage 1 + 2: $n_{DM} = 178,691$, $n_{noDM} = 1,296,113$), we identified 32 novel eGFR loci ($P_{Joint}$ or $P_{DM}$ or $P_{noDM} < 5 \times 10^{-8}$, Fig. 3a, Table 2, Supplementary Data 5, Supplementary Fig. 7b): (i) 30 novel loci by the joint test; (ii) two additional loci by the noDM-only search (near *FAT4, SLC2A4*). While no locus was additionally identified by the DM-only search, two of the 32 novel loci were genome-wide significant in DM ($P_{DM} < 5 \times 10^{-8}$; near *SH3BP4, LOXL4;* also identified by joint test).

These 32 novel loci included the three replicated novel loci from the discovery+replication approach (near *ATP12A, SERTAD2,* and *ABCC2*) (Table 2). The 29 additional novel loci included the *CUBN* locus, which is well-known for urinary albumin-to-creatinine ratio (UACR)[39] and microalbuminuria[40] (Fig. 4). The identified *CUBN* locus eGFR lead variant, rs11254238, showed (i) no correlation with the known signals for UACR ($r^2 < 0.001$, D' > 0.24 to any of the known independent signal variants for UACR, rs141493439, rs45551835, rs557338857, rs562661763, rs74375025), (ii) no effect on UACR[41] ($P = 0.57$, Supplementary Data 6), and (iii) a twice as large effect on eGFR in DM compared to noDM, but the difference was not significant ($b_{DM} = 0.0048$, $b_{noDM} = 0.0021$, $P_{Diff} = 0.07$).

**Table 1 Search for difference loci identified seven eGFR loci with established difference between DM and noDM.**

| Locus | rsid | Chr:Pos | ea | Stage 1 (UKB, CKDGen) | | | | | | Stage 2 (MVP, MGI, HUNT) | | | | | Combined stage |
| | | | | eaf | beta_DM | P_DM | beta_noDM | P_noDM | P_Diff | beta_DM | P_DM | beta_noDM | P_noDM | P_Diff,1-sided | P_Diff |
|---|---|---|---|---|---|---|---|---|---|---|---|---|---|---|---|
| *Discovery+replication design* | | | | | | | | | | | | | | | |
| *Difference test approach* | | | | | | | | | | | | | | | |
| [UMOD/PDILT] | rs77924615 | 16:20392332 | G | 0.79 | −0.019 | 2.3E-53 | −0.011 | 5.9E-188 | **1.2E-12** | −0.035 | 2.2E-62 | −0.015 | 2.6E-80 | **1.7E-20** | 1.3E-27 |
| *Overall+difference approach* | | | | | | | | | | | | | | | |
| [TPPP] | rs434215 | 5:699046 | A | 0.28 | −0.011 | 9.8E-10 | −0.004 | 1.2E-20 | **2.6E-05** | −0.013 | 7.3E-11 | −0.006 | 3.2E-13 | **3.2E-04** | 2.5E-09 |
| [MED1/NEUROD2] | rs55722796 | 17:37612086 | T | 0.76 | −0.001 | 0.28 | −0.006 | 5.3E-81 | **2.5E-05** | −8.0E-04 | 0.69 | −0.007 | 1.5E-20 | **0.004** | 1.1E-06 |
| *Combined stage design* | | | | | | | | | | | | | | | |
| *Difference test approach* | | | | | | | | | | | | | | | |
| [CSRNP1] | rs1828678 | 3:39195517 | G | 0.32 | −0.005 | 2.2E-05 | 4.0E-04 | 0.20 | 7.2E-06 | −0.005 | 0.009 | 3.0E-04 | 0.64 | 0.009 | **3.8E-08** |
| [DCDC5] | rs963837 | 11:30749090 | T | 0.59 | −0.009 | 3.3E-18 | −0.005 | 5.0E-76 | 1.9E-04 | −0.015 | 2.0E-19 | −0.006 | 5.0E-21 | 7.8E-08 | **5.6E-09** |
| [NRIP1] | rs1882963 | 21:16560118 | C | 0.21 | −0.008 | 3.3E-07 | −0.002 | 2.3E-09 | 8.9E-05 | −0.011 | 1.0E-07 | −0.004 | 2.0E-07 | 8.0E-04 | **3.8E-08** |
| *Overall+difference test approach* | | | | | | | | | | | | | | | |
| [SLC22A2] | rs2619264 | 6:160635258 | G | 0.21 | −0.005 | 2.1E-04 | −0.002 | 9.8E-07 | 0.013 | −0.009 | 8.6E-06 | −0.002 | 0.004 | 9.5E-04 | **4.5E-05** |

Shown are seven locus lead variants established with significant difference by two designs and two approaches using stage 1 (CKDGen and UKB, $n_{DM}$ = 109,993, $n_{noDM}$ = 1,070,999), stage 2 (MVP, MGI, and HUNT, $n_{DM}$ = 68,698; $n_{noDM}$ = 225,114), and combined data ($n_{DM}$ = 178,691, $n_{noDM}$ = 1,296,113). (i) Discovery+replication design: three variants discovered in stage 1 data were replicated in stage 2, including one by difference test approach (stage 1 $P_{Diff} < 5 \times 10^{-8}$ and stage 2 one-sided $P_{Diff} < 0.0125 = 0.05/4$, corrected for four variants discovered at stage 1) and two further variants by the overall+difference test approach (610 variants[16] with $P_{Overall} < 5 \times 10^{-8}$, stage 1 $P_{Diff} < 0.05/610$ and stage 2 one-sided $P_{Diff} < 0.05/4$). (ii) Combined stage design: four additional variants were identified in the combined stage design, including three by the difference test approach ($P_{Diff} < 5 \times 10^{-8}$) and one further by the overall+difference test approach (610 variants[16] with $P_{Overall} < 5 \times 10^{-8}$, stage 1 $P_{Diff} < 0.05/610$). Identifying P-values are stated in bold. Full summary statistics by DM status and stages are shown in Supplementary Data 2.
*DM* diabetes mellitus, *Chr* chromosome, *Pos* position (GRCh37), *ea* effect allele, *eaf* effect allele frequency in DM, Stage 1, *beta* Genetic effect estimates on log eGFR per allele, *p* Association P-value, $P_{Diff}$ Difference P-value.

Approximate conditional analyses (combined stage, European only, Methods) identified two additional variants independently associated with eGFR at two joint-test loci (near *SERTAD2* and *PIK3CG*, $P_{Joint\_cond} < 5 \times 10^{-8}$, Supplementary Table 2). This raised the total number of newly identified eGFR-associated variants to 34.

Among the 34 novel eGFR variants, four showed a suggestive DM/noDM-difference for eGFR ($P_{Diff} < 0.0015 = 0.05/34$, Table 2, Fig. 3b): one was a DM-only effect (near *SH3BP4*) and three showed more pronounced effects in DM (near *LOXL4*, *ALPL*, and *PIK3CG*). These patterns were consistent also in stage 1 and stage 2 separately (Supplementary Fig. 5). Of note, the difference test is mathematically dependent on the joint test and the stratified tests, but not on the overall test[24]. Therefore, the observed difference in the overall+difference test approach applied to the same data is an established significant difference, while the observed difference in loci established by the joint or stratified tests is termed here as suggestive difference.

In summary, the joint test and stratified tests identified 34 genome-wide significant independent eGFR variants across 32 novel eGFR loci. These included four loci with suggestive DM/noDM-difference, which yielded a total of 11 eGFR loci with established or suggestive DM/noDM-difference.

**Interaction with Hba1C and overlap with associations for DM risk, glycemic and other traits**. We explored the lead variants of the 11 eGFR loci identified with DM/noDM-difference (i.e., established/suggestive SNP-by-DM interaction) for SNP-by-HbA1c interaction in UK Biobank ($n = 368,005$), making full use of the continuous variable HbA1c instead of binary DM status. For all 11 variants, the SNP main effects and SNP-by-HbA1c interaction effect sizes were directionally consistent with main and interaction effects sizes in the SNP-by-DM interaction analysis (Supplementary Data 7, Methods). This underscored again the negative interaction effect for the 10 of the 11 variants with larger effects in DM or DM-only effects, while the one variant (near *MED1/NEUROD2*) showed a negative interaction effect for the SNP-by-DM as well as the SNP-by-HbA1c interaction.

We were also interested in whether any of the 11 loci with DM/noDM-difference overlapped with known genome-wide significant loci for type 1 DM[42], type 2 DM[19], or glycemic traits[43]. None of the 11 loci overlapped with type 1 DM, type 2 DM, or glycemic traits loci (all 11 lead variants $P > 5.0 \times 10^{-8}$ in published GWAS for DM risk, glucose, or insulin levels, Supplementary Data 8). None of the 11 variants was associated with type 1 DM judged at Bonferroni-corrected significance ($P > 0.05/11 = 4.5 \times 10^{-3}$). Three of the 11 variants were associated with type 2 DM ($P < 0.05/11 = 4.5 \times 10^{-3}$, rs77924615 near *UMOD-PDILT*, rs55722796 near *MED1-NEUROD2*) or fasting glucose (rs963837 near *DCDC5*). For all three variants, the eGFR-lowering allele was associated with decreased type 2 DM risk or fasting glucose. Thus, the SNP effect on eGFR cannot be fully explained by DM status, which would have yielded a DM risk increasing or glucose-increasing effect by the eGFR-lowering allele. The observation is in line with a pleiotropic effect on DM and eGFR, but with adverse effects for one of the two (eGFR or DM/glucose) and a beneficial effect for the other, which should be considered in drug design when applicable. When taking-into-account the DM/noDM-stratum where the eGFR effect was more pronounced, the three variants consisted of (i) two variants with stronger eGFR-effects in DM (still significant eGFR effect in noDM, near *UMOD/PDILT* and *DCDC5*) and (ii) one variant with a noDM-only effect on eGFR (near *MED1-NEUROD2*).

We also queried the Open Targets Genetics database[44] for associations of the 11 variants with other traits. We found 126

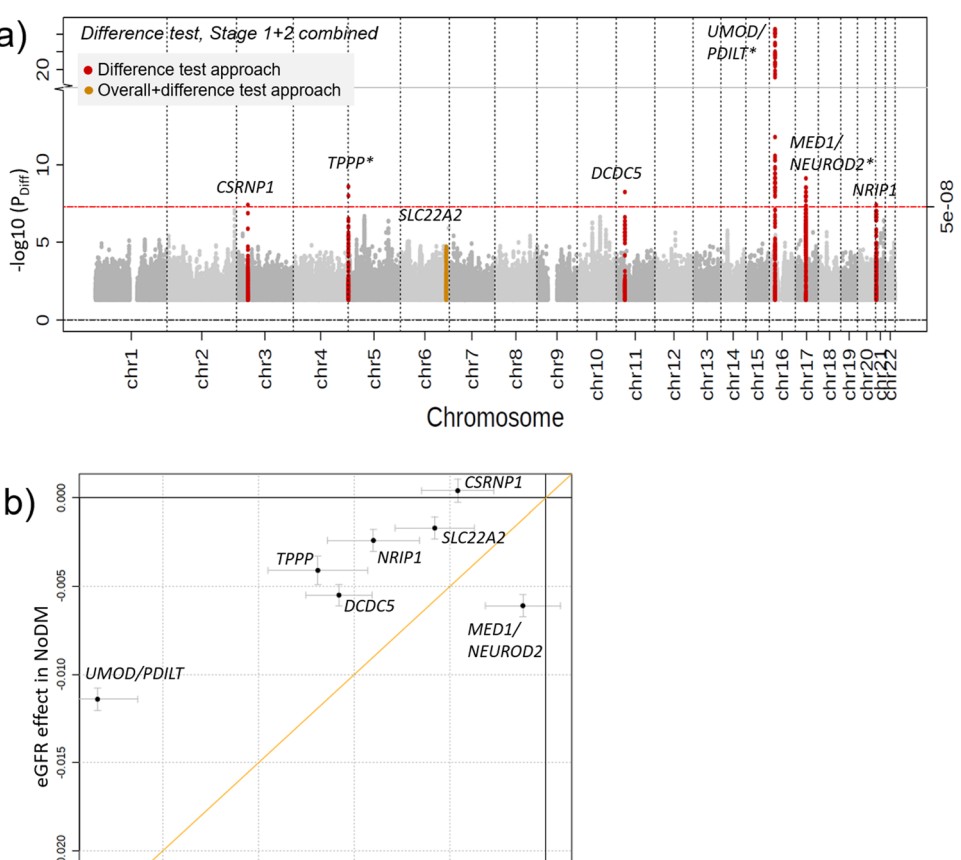

**Fig. 2 Seven eGFR loci with differential effects by diabetes status.** We searched for DM/noDM-differential genetic associations on eGFR using the difference test approach and the overall+difference approach in combined stage 1 + 2 (CKDGen, UKB, MVP, MGI, and HUNT; $n_{DM} = 178{,}691$; total $n_{noDM} = 1{,}296{,}113$). Seven difference loci were identified. **a** Shown are difference test P-values over chromosomal base position (Manhattan plot) highlighting the six loci identified by the difference test approach (red, $P_{Diff} < 5 \times 10^{-8}$) and the one locus identified by the overall+difference test approach (orange, 610 variants[16] with stage 1 $P_{Overall} < 5 \times 10^{-8}$, $P_{Diff} < 0.05/610 = 8.2 \times 10^{-5}$). Loci are annotated by the name(s) of the nearest gene(s); asterix indicates loci that were also identified by the discovery+replication design (Table 1, Supplementary Fig. 1). **b** Shown is a comparison of DM-/noDM-specific eGFR-effect sizes for the seven identified difference lead variants based on combined stage 1 + 2 data. Effect sizes are aligned to the eGFR-decreasing alleles in noDM except for CSRNP1 (aligned to eGFR-decreasing allele in DM). Error bars reflect 95% confidence intervals of the estimated genetic effect.

genome-wide significant associations ($P < 5 \times 10^{-8}$, Supplementary Data 9), particularly for hypertension and blood counts: for 3 of the 11 variants, the eGFR-decreasing alleles were associated with increased risk of hypertension (near UMOD/PDILT, DCDC5, and PIK3CG). The variant rs55722796 near MED1/NEUROD2, solely associated with eGFR in noDM individuals, was not associated with hypertension, but the eGFR-decreasing allele (in noDM) was associated with decreased blood counts (i.e., decreased red blood cells, hemoglobin, and hematocrit).

**GWAS for eGFR in individuals with DM identified 29 genome-wide significant loci.** Previous searches for SNP-effects on eGFR-based kidney function outcome in DM patients had identified eight genome-wide significant loci (i.e., eGFR-based DKD loci; four identified in type 1 DM patients[23], four other in type 1 or type 2 DM patients[22]). We wanted to understand the overlap of these with the genome-wide significant loci in our DM-only GWAS on eGFR ($n_{DM} = 178{,}691$, combined stage, Methods): we observed 29 genome-wide significant eGFR loci in DM ($P_{DM} < 5 \times 10^{-8}$, Fig. 5, Supplementary Data 10). These included: (i) 27 novel eGFR loci in DM, and (ii) two of the four previously

identified DKD loci among type 1 or type 2 DM patients[22] (near UMOD/PDILT and PRKAG2; $r^2 = 0.60$ and 0.94, respectively, between eGFR and DKD lead variants). None of the previously identified DKD loci among type 1 DM patients[23] were identified. Two of the 27 novel eGFR loci in DM were detected here as completely novel eGFR loci (near SH3BP4 and LOXL4; also detected by the joint test). The 29 identified eGFR loci in DM included 6 of our 11 difference loci (near UMOD/PDILT, TPPP, DCDC5, SH3BP4, NRIP1, and LOXL4).

**In silico evidence prioritized 18 genes by protein- or expression-altering variants or as human kidney disease monogene.** Genes at GWAS loci for eGFR might pinpoint relevant drug targets for kidney function[18]. Dissecting eGFR loci as DM-only, noDM-only, or shared will help define the target population for potential therapy. Gene PrioritiSation (GPS) was conducted previously[16] for all genes underneath the 424 eGFR-associated loci in a meta-analysis of CKDGen and UKB (i.e., our stage 1 studies) focused on European-ancestry. Since all 7 difference loci were among these 424 loci, we used this established GPS tool to extract in-silico functional evidence for the 159 genes

**Table 2 Allowing for DM/noDM difference identified 32 novel eGFR loci.**

| Locus | rsid | Source | ea | eaf | Stage 1 | | | | | Stage 2 | | | | | Combined stage | | | | |
|---|---|---|---|---|---|---|---|---|---|---|---|---|---|---|---|---|---|---|---|
| | | | | | beta$_{DM}$ | $P_{DM}$ | beta$_{noDM}$ | $P_{noDM}$ | $P_{joint}$ | beta$_{DM}$ | $P_{DM}$ | beta$_{noDM}$ | $P_{noDM}$ | $P_{joint}$ | beta$_{noDM}$ | $P_{DM}$ | $P_{noDM}$ | $P_{Diff}$ | $P_{Joint}$ |
| *Discovery+replication design* | | | | | | | | | | | | | | | | | | | |
| [ATP12A] | rs7324815 | joint | C | 0.15 | −0.0048 | 0.0012 | −0.0016 | 1.6E-05 | 4.9E-08 | −0.006 | 0.0060 | −0.0028 | 4.7E-04 | 6.0E-05 | −0.0028 | 4.4E-05 | 6.0E-08 | 0.0088 | 7.0E-11 |
| [SERTAD2] | rs12614829 | joint/noDM | T | 0.19 | −7.0E-04 | 0.65 | −0.002 | 3.0E-08 | 8.2E-09 | −0.0112 | 8.7E-08 | −0.0041 | 2.5E-07 | 1.7E-12 | −0.0041 | 5.7E-04 | 2.0E-12 | 0.10 | 1.7E-14 |
| [ABCC2] | rs56080571 | joint/noDM | G | 0.06 | −3.0E-04 | 0.90 | −0.0037 | 5.4E-09 | 1.3E-07 | −0.0029 | 0.41 | −0.0056 | 1.9E-04 | 7.7E-04 | −0.0056 | 0.55 | 6.4E-10 | 0.22 | 3.5E-09 |
| *Combined stage design* | | | | | | | | | | | | | | | | | | | |
| *with suggestive difference* | | | | | | | | | | | | | | | | | | | |
| [SH3BP4] | rs4463171 | joint/dm | A | 0.16 | −0.0075 | 2.4E-07 | −6.0E-04 | 0.12 | 3.4E-07 | −0.0063 | 0.01 | −0.0013 | 0.16 | 0.02 | −0.0013 | 8.8E-09 | 0.08 | 8.4E-08 | 7.0E-09 |
| [ALPL] | rs36053309 | joint | C | 0.79 | −0.0052 | 4.2E-05 | −0.001 | 0.0027 | 1.0E-06 | −0.0055 | 0.0066 | −3.0E-04 | 0.68 | 0.02 | −3.0E-04 | 5.6E-07 | 0.0059 | 1.2E-05 | 2.4E-08 |
| [LOXL4] | rs11189526 | joint/dm | G | 0.32 | −0.0028 | 0.0092 | −0.0012 | 8.7E-05 | 5.5E-05 | −0.008 | 3.6E-06 | −0.0017 | 0.0072 | 3.0E-07 | −0.0017 | 4.0E-06 | 2.3E-05 | 6.5E-04 | 2.7E-09 |
| [PIK3CG] | rs2392929 | joint/noDM | G | 0.19 | −0.0037 | 0.0050 | −0.0014 | 1.6E-04 | 2.2E-06 | −0.008 | 7.4E-05 | −0.0013 | 0.07 | 6.2E-05 | −0.0013 | 7.6E-06 | 1.6E-05 | 0.0011 | 3.5E-09 |
| *without suggestive difference* | | | | | | | | | | | | | | | | | | | |
| [DOCK7] | rs10789120 | joint | G | 0.71 | −0.0039 | 5.4E-04 | −0.0011 | 1.9E-04 | 8.2E-06 | −0.0043 | 0.02 | −0.002 | 0.0029 | 0.0011 | −0.002 | 3.1E-05 | 8.2E-05 | 0.0030 | 2.0E-08 |
| [EBF2] | rs13439370 | joint | C | 0.63 | −0.0049 | 7.8E-05 | −0.0014 | 1.2E-05 | 3.4E-08 | −0.003 | 0.13 | −0.0022 | 0.0027 | 0.0026 | −0.0022 | 3.2E-05 | 1.4E-06 | 0.0039 | 1.6E-09 |
| [KIF18A] | rs17310049 | joint | T | 0.82 | −0.0058 | 4.8E-04 | −0.0015 | 6.8E-05 | 5.0E-08 | −0.0025 | 0.24 | −0.0013 | 0.10 | 0.14 | −0.0013 | 4.6E-04 | 6.7E-06 | 0.02 | 4.5E-08 |
| [BACH1] | rs379592 | joint | G | 0.48 | −0.0029 | 0.0036 | −0.0011 | 8.1E-06 | 5.6E-08 | −0.0031 | 0.07 | −0.0016 | 0.0097 | 0.0058 | −0.0016 | 8.3E-04 | 1.6E-07 | 0.05 | 7.5E-10 |
| [CUBN] | rs11254238 | joint | C | 0.10 | −0.0042 | 0.02 | −0.0019 | 8.6E-05 | 4.3E-06 | −0.0068 | 0.02 | −0.0032 | 0.0032 | 0.0012 | −0.0032 | 0.0014 | 2.1E-06 | 0.07 | 4.4E-08 |
| [RASSF6] | rs17804499 | joint | C | 0.05 | −0.0064 | 0.04 | −0.0034 | 2.5E-07 | 1.7E-07 | −0.0081 | 0.04 | −0.0037 | 0.01 | 0.0026 | −0.0037 | 0.0026 | 6.8E-08 | 0.12 | 3.8E-09 |
| [TET2] | rs67149069 | joint/noDM | C | 0.83 | −6.0E-04 | 0.71 | −0.002 | 1.2E-07 | 8.5E-09 | 0.0013 | 0.55 | −0.0016 | 0.05 | 0.12 | −0.0016 | 0.96 | 2.7E-09 | 0.12 | 2.7E-08 |
| [FBXL17] | rs56855707 | joint/noDM | A | 0.15 | −0.0037 | 0.0092 | −0.002 | 1.8E-07 | 3.1E-10 | −0.0043 | 0.06 | −0.0025 | 0.0040 | 0.0040 | −0.0025 | 0.0016 | 5.6E-10 | 0.13 | 3.5E-12 |
| [NRBF2] | rs13095 | joint | G | 0.52 | −0.0026 | 0.0077 | −0.0011 | 6.5E-06 | 1.2E-07 | −0.0021 | 0.23 | −0.0016 | 0.01 | 0.01 | −0.0016 | 0.0052 | 1.6E-07 | 0.15 | 3.9E-09 |
| [TTC27] | rs6543664 | joint/noDM | G | 0.82 | −0.0029 | 0.03 | −0.0019 | 4.9E-07 | 4.9E-09 | −0.0048 | 0.04 | −0.0016 | 0.06 | 0.02 | −0.0016 | 0.0037 | 1.3E-08 | 0.17 | 2.7E-10 |
| [PSEN2] | rs1295641 | joint/noDM | T | 0.50 | −1.0E-04 | 0.93 | −0.0012 | 1.8E-06 | 3.0E-07 | −7.0E-04 | 0.66 | −0.0024 | 1.0E-04 | 3.6E-04 | −0.0024 | 0.75 | 1.4E-09 | 0.27 | 1.0E-08 |
| [REM2] | rs2295904 | joint | C | 0.17 | −0.0015 | 0.28 | −0.0016 | 1.4E-05 | 3.7E-06 | −0.0067 | 0.0022 | −0.0021 | 0.01 | 3.3E-04 | −0.0021 | 0.01 | 3.2E-07 | 0.28 | 3.0E-08 |
| [BCL6] | rs78158637 | joint/noDM | G | 0.83 | −0.001 | 0.54 | −0.0021 | 1.6E-08 | 1.1E-09 | 0 | 0.98 | −0.0021 | 0.01 | 0.03 | −0.0021 | 0.61 | 1.3E-10 | 0.28 | 5.0E-10 |
| [AUTS2] | rs3750170 | joint | A | 0.33 | −0.0022 | 0.06 | −0.0018 | 2.3E-07 | 6.0E-08 | −0.0055 | 0.02 | −0.0018 | 0.03 | 0.01 | −0.0018 | 0.0091 | 5.8E-08 | 0.32 | 5.6E-09 |
| [SLC2A2] | rs7630490 | joint/noDM | C | 0.84 | −0.0033 | 0.02 | −0.0017 | 1.5E-05 | 6.7E-08 | −8.0E-04 | 0.71 | −0.0017 | 4.3E-04 | 0.0015 | −0.0029 | 0.02 | 1.5E-09 | 0.47 | 1.5E-09 |
| [ZFP36L1] | rs72731564 | joint/noDM | T | 0.19 | −1.0E-04 | 0.93 | −0.0018 | 1.3E-06 | 3.1E-07 | −0.0026 | 0.22 | −0.0028 | 5.0E-04 | 0.0011 | −0.0028 | 0.43 | 2.8E-09 | 0.49 | 2.1E-08 |
| [POLD3] | rs1944933 | joint/noDM | T | 0.53 | −7.0E-04 | 0.50 | −0.0012 | 2.5E-06 | 2.4E-07 | −0.0011 | 0.51 | −0.0023 | 2.5E-04 | 6.0E-04 | −0.0023 | 0.36 | 2.0E-09 | 0.58 | 7.5E-09 |
| [EEF1DP3] | rs703214 | joint | C | 0.49 | −0.0022 | 0.03 | −0.0013 | 9.5E-08 | 2.1E-09 | −4.0E-04 | 0.80 | −0.0013 | 0.76 | 0.92 | −2.0E-04 | 0.05 | 1.1E-07 | 0.58 | 2.9E-08 |
| [DUSP6] | rs1472212 | joint/noDM | T | 0.83 | −0.0023 | 0.14 | −0.0017 | 6.1E-06 | 4.9E-07 | −0.0037 | 0.19 | −0.0017 | 9.1E-06 | 1.1E-05 | −0.0037 | 0.05 | 1.3E-09 | 0.68 | 5.1E-10 |
| [ARMC4] | rs7896951 | joint/noDM | A | 0.47 | 0 | 0.98 | −0.0012 | 3.4E-06 | 3.1E-07 | −0.0022 | 0.04 | −0.0022 | 3.6E-04 | 1.4E-04 | −0.0022 | 0.26 | 2.7E-09 | 0.74 | 6.0E-09 |
| [SLC25A15] | rs2282026 | joint/noDM | G | 0.46 | −0.0011 | 0.25 | −0.0015 | 6.8E-09 | 3.7E-11 | −0.0029 | 0.08 | −9.0E-04 | 0.13 | 0.08 | −9.0E-04 | 0.08 | 9.9E-11 | 0.82 | 1.3E-10 |
| [MSC] | rs13281719 | joint/noDM | C | 0.91 | −0.0018 | 0.42 | −0.0027 | 7.2E-08 | 3.8E-06 | −0.0041 | 0.15 | −0.0046 | 2.7E-05 | 7.0E-05 | −0.0046 | 0.14 | 5.7E-09 | 0.82 | 1.9E-08 |
| [LIMCH1] | rs75228450 | joint | A | 0.87 | −0.0024 | 0.13 | −0.0023 | 2.5E-06 | 3.5E-07 | −0.0034 | 0.26 | −0.0027 | 0.01 | 0.03 | −0.0027 | 0.07 | 7.5E-08 | 0.89 | 3.1E-08 |
| [SLC2A4] | rs117643180 | noDM | A | 0.02 | 0.0051 | 0.16 | −0.0053 | 5.6E-06 | 4.3E-09 | 0.0018 | 0.73 | −0.002 | 0.32 | 0.58 | −0.002 | 0.19 | 3.1E-08 | 0.0030 | 6.6E-08 |
| [FAT4] | rs1506363 | noDM | T | 0.83 | 0.0015 | 0.36 | −0.0019 | 4.6E-07 | 3.6E-08 | −0.0037 | 0.08 | −0.0014 | 0.09 | 0.05 | −0.0014 | 0.73 | 2.2E-08 | 0.28 | 1.6E-07 |

Shown are 32 novel variants identified with genome-wide significant eGFR association allowing for DM/noDM-difference by two designs and two approaches (joint or stratified test approaches) using stage 1 ($n_{DM} = 109,993$, $n_{noDM} = 1,070,999$), stage 2 ($n_{DM} = 68,698$; $n_{noDM} = 225,114$), or combined data ($n_{DM} = 178,691$, $n_{noDM} = 1,296,113$). (i) Discovery+replication design: 3 variants with $P_{Joint}$, $P_{DM}$, or $P_{noDM} <5 \times 10^{-8}$ in stage 1 were replicated in stage 2 ($P_{Joint}$, $P_{DM}$, or $P_{noDM} <0.05/25 = 0.002$). (ii) Combined stage design: 29 additional variants were identified with $P_{Joint}$, $P_{DM}$, or $P_{noDM} <5 \times 10^{-8}$ in stage 1+2 (4 with suggestive difference, $P_{diff} <0.05/32$). Details are shown in Supplementary Data 6. Source is the identifying test.

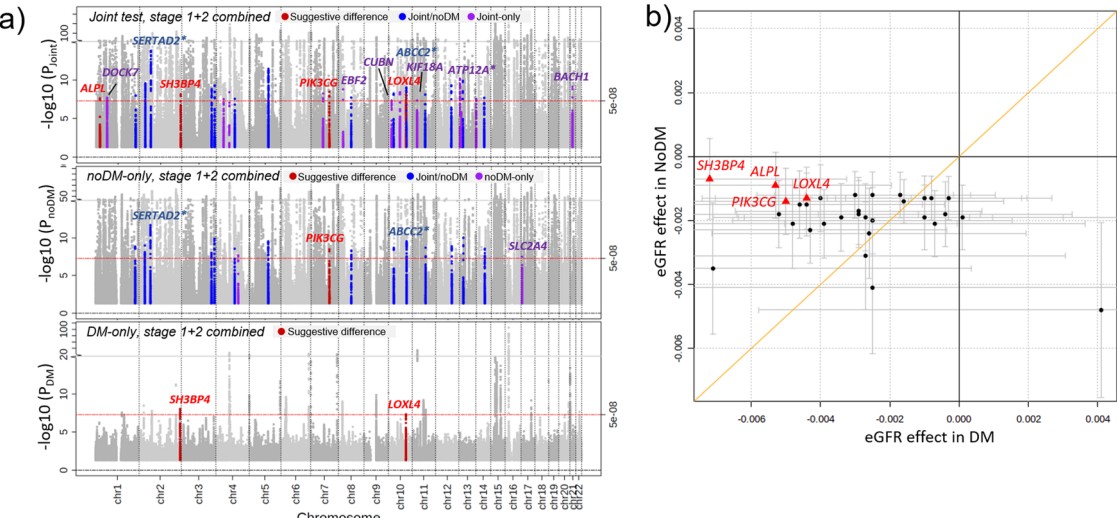

**Fig. 3 Accounting for potential DM-/noDM-differences identified 32 novel eGFR loci.** We searched for novel loci associated with eGFR allowing for DM-/noDM-difference using the joint test or DM-/noDM-stratified tests approaches in combined stage $1+2$ ($n_{DM} = 178,691$; total $n_{noDM} = 1,296,113$). We found 32 novel genome-wide significant eGFR loci ($P < 5 \times 10^{-8}$, >500 kB distant of known eGFR loci compared to previous work[16,20]): 30 by joint, 17 by noDM-only and 2 by DM-only test. **a** Shown are $P$-values for eGFR based on joint, noDM-only, and DM-only test over chromosomal position. Highlighted in red are loci with suggestive DM/noDM-difference ($P_{Diff} < 0.05/34$; corrected for 34 independent variants across 32 loci), blue for loci identified by joint and noDM-only test (15 loci), and purple for loci that were only identified by joint test (upper panel) or noDM-only test (middle panel). Loci were annotated by nearest genes if $P_{Diff} < 0.10$ or if they were also identified by the discovery+replication design (the latter also indicated by asterix, Table 2). **b** Shown is a comparison of DM-/noDM-specific eGFR-effect sizes for the 32 novel eGFR locus lead variants. Highlighted in red are the locus names of loci with suggestive DM/noDM-difference ($P_{Diff} < 0.05/34$; corrected for 34 independent variants across 32 loci). Effect sizes are aligned to the eGFR-decreasing alleles in noDM. Error bars reflect 95% confidence intervals of the estimated genetic effect.

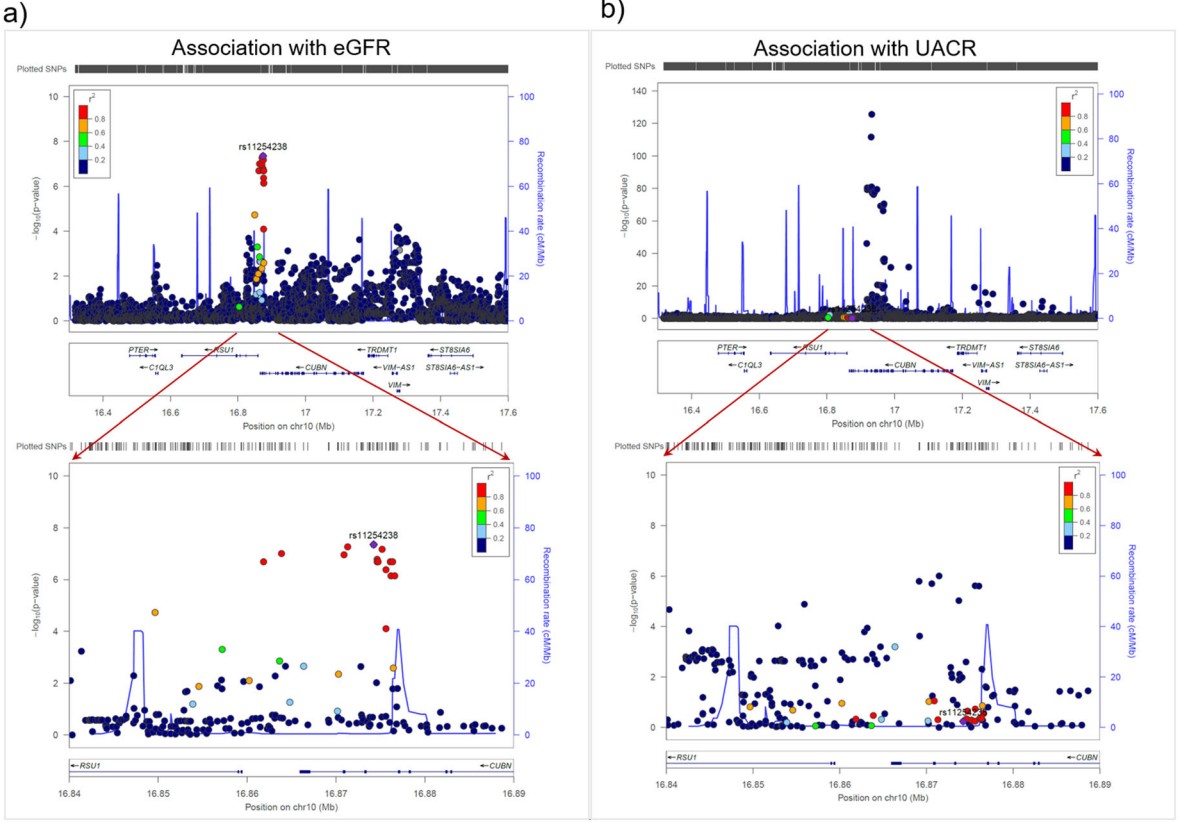

**Fig. 4 The variants associated with eGFR in the *CUBN* locus differ from those associated with urinary albumin-to-creatinine ratio.** Shown are $P$-values for associations at the wider (top) and more narrow (bottom) *CUBN* locus region for **a** eGFR (joint test $P$-values, $n_{DM} = 178,691$ and $n_{noDM} = 1,296,113$) and **b** urinary albumin-to-creatinine ratio (UACR; $P$-values from ref. [39], $n = 564,257$). Lead variant for eGFR is rs11254238; color codes variants' correlation $r^2$ to rs11254238 in all panels.

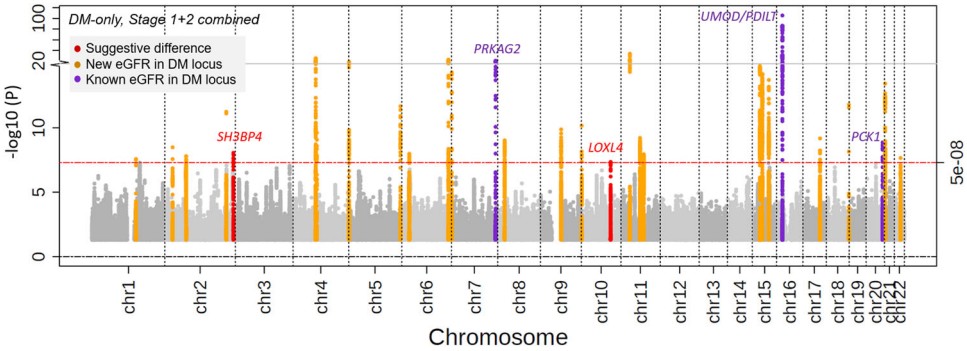

**Fig. 5 DM-only eGFR GWAS identified 29 loci, including 27 novel for eGFR in DM.** Shown are eGFR association *P*-values in individuals with DM over chromosomal position in combined stage ($n_{DM} = 178,691$). This DM-specific analysis identified 29 independent eGFR-associated loci in DM. Compared to known DKD loci[22,23] (i.e., association with eGFR or CKD in type 1 and/or type 2 DM individuals) and known overall eGFR loci[16,20], 2 loci are novel for eGFR overall and novel for DKD (red), 24 are novel for DKD but known for eGFR (orange), and 3 are known DKD and known eGFR loci (purple).

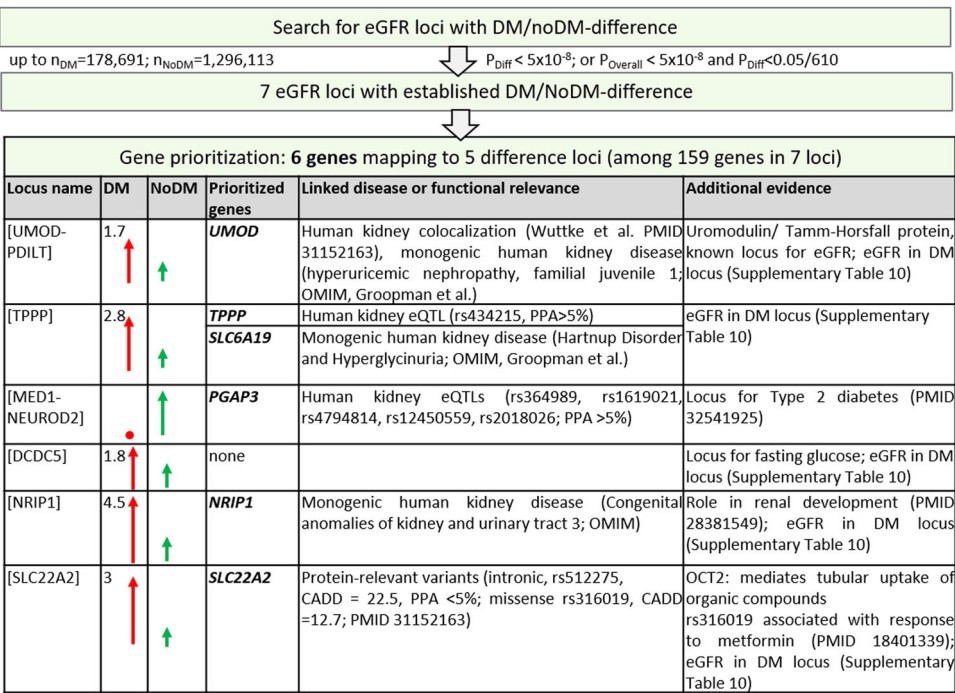

| Locus name | DM | NoDM | Prioritized genes | Linked disease or functional relevance | Additional evidence |
|---|---|---|---|---|---|
| [UMOD-PDILT] | 1.7 ↑ | ↑ | **UMOD** | Human kidney colocalization (Wuttke et al. PMID 31152163), monogenic human kidney disease (hyperuricemic nephropathy, familial juvenile 1; OMIM, Groopman et al.) | Uromodulin/ Tamm-Horsfall protein, known locus for eGFR; eGFR in DM locus (Supplementary Table 10) |
| [TPPP] | 2.8 ↑ | ↑ | **TPPP** | Human kidney eQTL (rs434215, PPA>5%) | eGFR in DM locus (Supplementary Table 10) |
| | | | **SLC6A19** | Monogenic human kidney disease (Hartnup Disorder and Hyperglycinuria; OMIM, Groopman et al.) | |
| [MED1-NEUROD2] | ● | ↑ | **PGAP3** | Human kidney eQTLs (rs364989, rs1619021, rs4794814, rs12450559, rs2018026; PPA >5%) | Locus for Type 2 diabetes (PMID 32541925) |
| [DCDC5] | 1.8 ↑ | ↑ | none | | Locus for fasting glucose; eGFR in DM locus (Supplementary Table 10) |
| [NRIP1] | 4.5 ↑ | ↑ | **NRIP1** | Monogenic human kidney disease (Congenital anomalies of kidney and urinary tract 3; OMIM) | Role in renal development (PMID 28381549); eGFR in DM locus (Supplementary Table 10) |
| [SLC22A2] | 3 ↑ | ↑ | **SLC22A2** | Protein-relevant variants (intronic, rs512275, CADD = 22.5, PPA <5%; missense rs316019, CADD =12.7; PMID 31152163) | OCT2: mediates tubular uptake of organic compounds rs316019 associated with response to metformin (PMID 18401339); eGFR in DM locus (Supplementary Table 10) |

**Fig. 6 Gene prioritization highlights six genes at loci with established DM/noDM-difference.** Shown are gene prioritization results for the seven loci with established difference (Table 1, Supplementary Fig. 8). We highlighted six genes based on association-driving variants (PPA > 5%) that were deleteriously protein-relevant or expression-modulating, genes that were known as human kidney monogenes (OMIM or ref. [78] with subsequent manual curation) and in addition *SLC22A2* due to its known link to metformin response.

at the 7 difference loci: here, we prioritized a gene when it mapped to an association-driving variant (i.e., 99% credible set variant with posterior probability of association ≥5%, Methods) that was relevant (i) for the protein (with high predicted deleteriousness[45], i.e., CADD score ≥15) or (ii) for expression in kidney tissue (eQTL, false-discovery rate ≥5%)[46,47] or (iii) when the gene was a known kidney disease monogene in human (Methods). We found 6 prioritized genes (Fig. 6, Supplementary Fig. 8): (i) *PGAP3* mapping to the locus with noDM-only effect on eGFR (near *MED1/NEUROD2*) and (ii) five genes in loci with more pronounced eGFR-effects in DM but direction-consistent effects in noDM (*TPPP, UMOD, SLC6A19, NRIP1,* and *SLC22A2*).

We also used the same approach as in the previous GPS tool[16] to prioritize the genes underneath the 32 novel eGFR loci (34 independent signals, Supplementary Data 11–15, Supplementary Table 3, Supplementary Fig. 9): (i) among the 40 genes in the four novel loci with suggestive DM/noDM-difference, 2 genes were human kidney disease monogenes (*HPS1* and *HPSE2*, Fig. 7a); (ii) for the 341 genes in the other 28 novel eGFR loci (i.e., noDM/ noDM-difference), 10 genes were prioritized (Fig. 7b): 6 genes contained an association-driving variant that was protein-deleterious (*AUTS2, CUBN, DVL2, RASSF6, SLC2A4,* and *ZFP36L1*), one gene mapped to an eQTL in glomerular tissue (*TNIK*), and 3 additional genes were human kidney disease monogenes (*SLC2A2, SLC30A9,* and *SLC7A7*). Particularly interesting was an association-driving variant, rs1801232, in *CUBN* that was a missense variant ($r^2 = 0.73$ to eGFR lead variant rs11254238). The missense variant rs17804499 in *RASSF6* had a particularly high probability of being the association-driving variant (posterior probability of association = 86%), which rendered this variant and gene a compelling candidate for

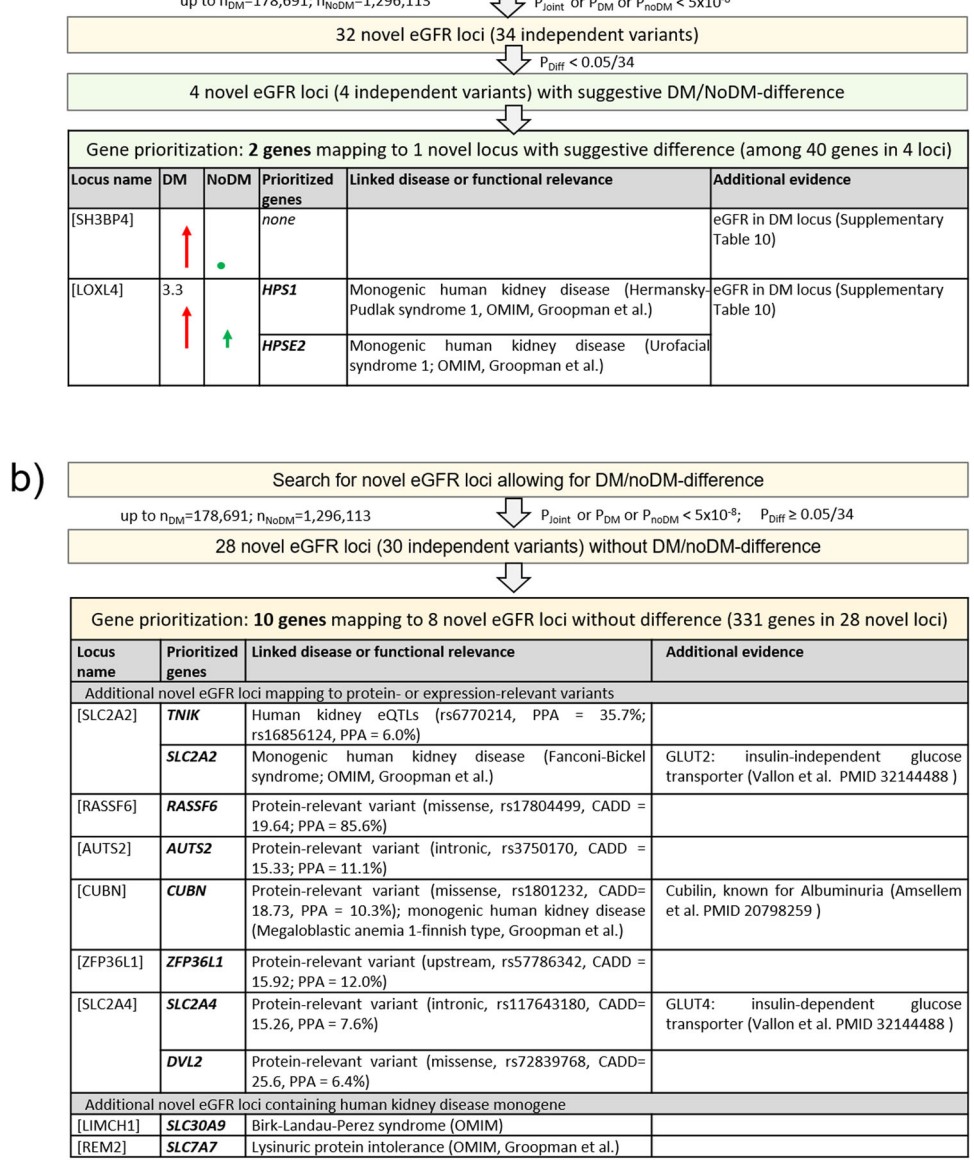

**Fig. 7 Gene prioritization highlights 12 genes at novel eGFR loci.** Shown are gene prioritization results for the 32 novel eGFR loci (Table 2, Supplementary Fig. 9): **a** for the four novel eGFR loci with suggestive difference, and **b** for the 28 other novel eGFR loci. We highlighted 12 genes based on association-driving variants (PPA > 5%) that were deleteriously protein-relevant or expression-modulating or genes that were known as human kidney monogenes (OMIM or ref. [78] with subsequent manual curation).

functional follow-up. Two genes, *SLC2A2* and *SLC2A4*, encode the glucose transporters GLUT2 and GLUT4 and reside in two novel eGFR loci.

In summary, across the identified seven loci identified with established or suggestive DM-/noDM-difference in eGFR-effects and the 32 novel eGFR loci, we prioritized 18 genes with evidence of protein-deleterious or expression-modulating variants or being human kidney disease monogenes. These genes are compelling candidates for functional follow-up and drug development pipelines.

**DM/noDM-stratified weights for GRS show similar explained variance as overall weights**. We assessed whether the GRS for eGFR, computed as an effect size weighted sum of eGFR-lowering alleles, could be improved by using DM/noDM-specific eGFR-

effects as weights rather than overall eGFR-effects. We built the GRS based on the previously identified 634 eGFR variants[16] (i.e., identified in stage 1 studies) or based on the seven variants with established DM/noDM-difference using different weighting schemes (overall effects, DM-/noDM-specific effects, or a combination; all weights from stage 1 data). We computed the GRS for each individual in the stage 2 study HUNT (unrelated, European-ancestry; $n_{DM} = 3023$, $n_{noDM} = 25,484$, Fig. 1d, Methods).

Across all weighting schemes, we found no significant DM/noDM-difference in the 634-variant GRS association with eGFR ($P_{Diff} > 0.05$, Table 3). The 634-variant GRS explained more of the eGFR variance in noDM compared to DM (e.g., $R^2 = 6.0\%$ vs. 4.0% using overall-effect weights). Since the absolute 634-variant GRS effect was similar in DM and noDM (e.g., beta per $sd_{GRS} = -2.54$ versus $-2.84$ ml/min/1.73 m² using overall-effect weights), this larger relative GRS effect in noDM can be attributed

**Table 3 Genetic risk score (GRS) association with eGFR and explained variance of eGFR separately for DM and noDM individuals.**

| GRS | DM $sd_Y = 13.4$ ml/min per 1.73m², $n = 3023$ | | | | noDM $sd_Y = 11.8$ ml/min per 1.73m², $n = 25,484$ | | | | $P_{diff}$ |
|---|---|---|---|---|---|---|---|---|---|
| | beta per $sd_{grs}$ | se per $sd_{grs}$ | $P$ | $R^2$ | beta per $sd_{grs}$ | se per $sd_{grs}$ | $P$ | $R^2$ | |
| GRS based on 634 overall eGFR variants | | | | | | | | | |
| Weighted by overall effects | −2.54 | 0.240 | 1.1E-25 | 4.0% | −2.84 | 0.072 | 7.6E-331 | 6.0% | 0.23 |
| Weighted by DM-specific effects | −2.21 | 0.241 | 1.2E-19 | 3.1% | −2.38 | 0.073 | 1.6E-231 | 4.4% | 0.49 |
| Weighted by noDM-specific effects | −2.50 | 0.240 | 5.8E-25 | 3.9% | −2.85 | 0.072 | 3.5E-355 | 6.0% | 0.16 |
| Weighted by overall or DM/noDM-specific effects[a] | −2.58 | 0.240 | 1.6E-26 | 4.1% | −2.84 | 0.072 | 7.6E-331 | 6.0% | 0.31 |
| GRS based on 7 difference eGFR variants | | | | | | | | | |
| Weighted by overall effects | −1.02 | 0.244 | 2.7E-05 | 0.98% | −0.68 | 0.074 | 6.9E-20 | 0.62% | 0.17 |
| Weighted by DM-specific effects | −1.04 | 0.244 | 2.2E-05 | 1.00% | −0.53 | 0.074 | 1.1E-12 | 0.49% | 0.045 |
| Weighted by noDM-specific effects | −1.02 | 0.244 | 3.2E-05 | 0.97% | −0.68 | 0.074 | 4.1E-20 | 0.62% | 0.19 |

Shown are results from GRS analyses separately for DM and noDM individuals. The GRS's were based on the previously established 634 independent eGFR index variants[16] (stage 1 studies) or on the 7 variants with established difference and computed in unrelated individuals of the HUNT study ($n_{DM} = 3023$, $n_{noDM} = 25,484$, one of the stage 2 studies). Three versions of each GRS were computed for each individual: GRS's weighted by overall genetic eGFR-effects, weighted by DM-specific effects or weighted by noDM-specific genetic eGFR-effects. For the GRS based on the 634 variants, a further GRS was weighted by overall effects for variants without observed DM/noDM-difference and weighted by DM-/noDM-specific effects for variants with observed difference. The association of the GRS on eGFR and the variance explained ($R^2$) were derived for DM and noDM separately via linear regression with the respective GRS as covariate and eGFR as outcome (not log-transformed, adjusted for age, sex, and principal components, Methods). Also shown is a $P$-value testing the GRS effect on eGFR for difference between DM compared to noDM.
*sd* standard deviation, *se* standard error, *beta* change in eGFR [ml/min/1.73 m²] per standard deviation of the GRSFIGURES.
[a]Using overall weights for loci without observed difference and DM- or noDM-specific weights for loci with significant difference for DM or noDM individuals, respectively.

—at least in part—to the smaller eGFR variance in noDM versus DM (HUNT standard deviation = 11.8 versus 13.4 ml/min/1.73 m², respectively). Yet, the 7-variant GRS explained more of the eGFR variance in DM compared to noDM (e.g., $R^2 = 0.98\%$ vs. 0.62% using overall-effect weights). The reason for this is that six of the seven variants had larger eGFR-effects in DM than in noDM, which accumulated to a larger absolute GRS effect in DM (e.g., beta per $sd_{GRS} = -1.02$ in DM compared to −0.68 in noDM using overall-effect weights).

When comparing different weighting schemes, we found no notable improvement in $R^2$ values by using DM-/noDM-specific weights compared to overall weights (Table 3). In the 634-variant GRS, using DM-specific weights even reduced $R^2$ in DM individuals ($R^2 = 3.1\%$ versus 4.0%). This may be attributable to the larger uncertainty in DM-specific weights estimated from the meta-analysis restricted to DM individuals. For the 7-variant GRS, DM-specific weights slightly increased $R^2$ in DM individuals, but not markedly ($R^2 = 1.0\%$ versus 0.98%). These findings underscore a similar performance of overall weights and DM-/noDM-specific weights when building a GRS for eGFR. The comparison of the 634-variant with the 7-variant GRS suggests that more differential effect loci and higher precision in DM-specific weights might substantially improve the genetically explained variance also in DM individuals.

## Discussion

In this GWAS on eGFR in ~180,000 individuals with DM and ~1.3 million individuals without DM, we established seven loci with significant DM/noDM-differential effects on eGFR near *UMOD/PDILT, TPPP, MED1/NEUROD2, CSRNP1, DCDC5, NRIP1*, and *SLC22A2*. We also identified 32 novel eGFR-associated loci when allowing for potential difference, including four loci with a suggestive difference near *SH3BP4, LOXL4, ALPL,* and *PIK3CG*. The 11 loci with established or suggestive difference included two loci with DM-only effects on eGFR (near *CSRNP1, SH3BP4*), one with noDM-only effect (near *MED1/NEUROD2*), and all others showed more pronounced effects in DM compared to noDM. Our GWAS focused on DM individuals confirmed two known DKD loci, but also identified 27 novel loci for eGFR in

DM, which make these potential new DKD loci. The DM-/noDM-stratified GRS analyses showed no improvement in explained trait variance when using DM-/noDM-stratified weights, but the seven identified DM/noDM-differential eGFR-effects explained more of the eGFR variance in DM individuals than in noDM individuals.

Our gene prioritization at the 11 loci with DM/noDM-difference and at the 28 other novel eGFR loci highlighted 18 genes: 10 genes mapped to eGFR-association-driving variants that were deleteriously protein-relevant or eQTLs in kidney tissue, which suggests them as potential drug targets to alter eGFR. Further eight genes were prioritized because they were known as human kidney disease monogenes, which made them plausible causal genes for these common variant findings. Particularly interesting was *CUBN* in a novel eGFR locus. The *CUBN* locus is the GWAS locus with the strongest effect on UACR[39] and microalbuminuria[40] and identified here for eGFR at genome-wide significance, but respective lead variants for eGFR and UACR are uncorrelated. The lead variant for eGFR, rs11254238, is highly correlated with a variant that alters the encoded protein, cubilin (rs1801232, $r^2 = 0.91$). We observed a twice as large effect on eGFR in DM compared to noDM in both discovery and replication data, but not statistically significantly different. This is in line with previously observed larger *CUBN* variants' effects on microalbuminuria[40] and UACR[41,48] in DM, which provides further evidence for an interaction of *CUBN* with DM status on kidney function. In fact, our previous sequencing of the *CUBN* gene found rare variants in *CUBN* that were associated with higher UACR and with better eGFR[49]. Our study now reports such a parallel association between UACR and eGFR now also for a common variant at the genome-wide significance levels. Together, our results provide further support for the importance to assess the physiological role of *CUBN* by functional studies not only for microalbuminuria, but also for finding pathophysiological explanations related to impaired filtration rate.

The *RASSF6* gene has been studied in relation to the kidney as well: *RASSF6* mediates apoptosis in various cells[50]. It is shown to be expressed in the slit diaphragm in glomeruli and the apical membranes in proximal renal tubular epithelial cells of rat

kidney. The same animal study suggests the involvement of the *RASSF6* pathway in contrast-induced nephropathy. Thus, the novel detection of the *RASSF6* locus for eGFR here and the lead variant, rs17804499, being protein-altering and with 86% probability causal render *RASSF6* a compelling candidate for the functional follow-up to help understand its role in kidney function that is elusive so far.

Another interesting gene was *PGAP3* in the *MED1-NEUROD2* locus, which mapped to an expression-modulating variant that was associated with eGFR in noDM, but not in DM individuals. *PGAP3* is a known gene for CKD and eGFR: eQTL analyses showed colocalization of the association signal with *PGAP3* expression[51]. *PGAP3* knockout mice developed larger glomeruli with deposition of immunoglobulins[52], although we acknowledge that this does not provide a mechanistic explanation for its association with eGFR. Eventually, if causal genes at eGFR loci with significant differences by DM status will be advanced as drug targets for reno-protective therapies, the knowledge about DM- or noDM specificity of the eGFR association can help define the target group for potential subsequent therapy, as DM-only, noDM-only, or both.

The difference in SNP effects between DM and noDM can be interpreted as SNP-by-DM-status interaction effect. To our knowledge, our study is the first genome-wide search for SNP-by-DM-status interaction effects for eGFR. Sensitivity analyses indicated that the observed differences were neither biased from the stratified modeling, nor from log-transformed eGFR, and the SNP-by-DM-status interaction was not explained by SNP-interaction with age, sex, hypertension, or body-mass-index[53]. The seven loci with an established difference are known loci for eGFR[16,20]. To our knowledge, the significant DM/noDM-differential effects for eGFR or eGFR-based CKD were identified here for the first time, except for the *UMOD-PDILT* locus[21]. At *UMOD-PDILT*, a well-known kidney function locus[16,20], the two-fold higher eGFR effect in DM compared to noDM confirmed previous observations for eGFR[21,54] and CKD[55]. Very interesting was the *TPPP* locus, where the three-fold higher eGFR effect in DM now draws attention to a locus that was just one of the hundreds of small effects eGFR loci. The lead variant for the difference, rs434215, modulates *TPPP* expression in tubulo-interstitial tissue[46], but the role of *TPPP* in diabetes and kidney disease is yet unknown. Previous studies on *TPPP* focused on the nervous system: *TPPP* is highly expressed in the brain and shown to affect neural microtubules[56].

Our GWAS on eGFR in DM provides a link to the genetics of DKD. Previous GWAS for DKD analyzed eGFR or eGFR-based CKD in type 1 and/or type 2 DM individuals[22,23]. Our GWAS in >178,000 DM individuals was 4- to 7-fold larger. These DM individuals were mostly from population-based studies of adult individuals. As such, the proportion of type 1 DM among DM individuals analyzed here reflects the proportion of type 1 DM of 5–10% among adult DM individuals[57]. Due to the substantially larger number of type 2 DM individuals, the identified DM/noDM-differential effects on eGFR may mostly reflect differences between type 2 DM versus noDM. A better distinction by type of DM would require more granular data focusing on the distinction between these two major diabetes groups and substantially larger data on type 1 DM. While the use of self-reports or one-time measurements of glucose or HbA1c to define DM here is typical also for GWAS on DM[19,58], this implies heterogeneity in the DM definition and may include some individuals without clinically manifest DM.

Yet, our 29 loci with genome-wide significant association with eGFR in DM confirmed two of eight previously identified DKD loci[22,23] and the 27 other are compelling new candidate loci for further analyses in studies specialized on DKD patients. A large sample size eGFR GWAS with a more heterogeneous spectrum of DM can be a powerful complementary approach to focused searches in DKD patients[22,23]. Several prioritized genes mapped to these novel eGFR loci in DM: *TPPP* already mentioned above and *SLC6A19*, *NRIP1*, *HPS1* as well as *HPSE2* with strong monogenic impact on kidney. All these genes reside in loci with more pronounced eGFR-effects in DM compared to noDM, but none of these effects was DM-only.

Our GWAS allowing for DM/noDM-differential effects identified 32 novel eGFR-associated loci compared to previous GWAS[16,20]. Some of our novel loci might be identified due to increased power by ~20% increased sample size or due to chance by using alternative statistical tests. However, four of the novel loci showed suggestive DM/noDM-difference, suggesting that their identification was facilitated by using tests allowing for differential effects[59]. The four suggestive difference loci included one locus, near *SH3BP4*, with eGFR effect only in DM.

In contrast to hundreds of loci found in GWAS for eGFR, the eGFR loci with significant DM/noDM-difference were few. This suggests a largely shared genetics of kidney function between DM and noDM individuals, which has an important implication for drug development: most drug interventions aimed at altering eGFR should thus be effective among persons with and without DM. This is mirroring what is observed for SGLT2 inhibitors—a medication originally developed for individuals with DM that is now also being tested for reno-protection among individuals without DM[13]. While our GWAS here is, to date, the largest for eGFR in DM individuals, we still might have missed loci with effects in DM-only. Power for interaction effects is generally smaller than for overall effects[60], and particularly reduced when one subgroup, like DM, is substantially smaller (~10%) than the other[28]. Future work with an increased sample size might detect more eGFR loci with differences by DM status, especially those with effects in DM-only. This GWAS included mostly European individuals and our findings require replication in non-European ancestries[61], particularly because of the large differences in DM prevalence across ancestries[62].

In summary, our results highlight the existence of DM- and noDM-specific genetic effects on kidney function, but emphasize that the majority of eGFR locus associations do not differ between individuals with and without DM. Larger DM-/noDM-stratified data on eGFR in the future will improve the detectability of differential effect loci and the precision of DM-specific weights. This might also improve DM-/noDM-stratified GRS prediction of eGFR. The identified eGFR loci with difference between DM- and noDM individuals include loci with effects only in DM as well as loci with effects in noDM. This has highly relevant implications, if the respective genes are advanced as a drug target: the specificity of the association might help sharpen the target group for such potentially arising drug therapies.

## Methods

### Definition of the outcome eGFR and study-specific participant information.
GFR was estimated in all study participants in all studies using serum creatinine measurements via the Chronic Kidney Disease Epidemiology Collaboration (CKD-EPI) formula[20]. Study-specific information on the utilized assay and year of measurement is given in Supplementary Data 1. We used the R package nephro[63], winsorized at 15 and 200 ml min$^{-1}$per 1.73 m$^2$, and logarithmized using a natural logarithm. For a better interpretation of effect size, we there used eGFR on the original scale in GRS analyses. Study-specific information on study design, sample size, sex and age, utilized serum creatinine assay, year of measurement, and average eGFR is given in Supplementary Data 1.

Each study is conducted according to the declaration of Helsinki; local ethics committees approved research protocols and participants provided written informed consent.

### Definition of DM status.
DM of each study participant at the time point of the serum creatinine measurement was defined either (i) as fasting plasma glucose ≥126 mg/dl (7.0 mmol/L) or diabetes therapy, or (ii) (fasting glucose unavailable)

as non-fasting plasma glucose ≥200 mg/dl (11.0 mmol/L) or diabetes therapy, or (iii) (glucose unavailable) as self-reported diabetes. For UKB, DM was defined as HbA1c≥48 mmol/mol (≥6.5%) or diabetes therapy (i.e., A10 ATC codes obtained from[64]).

**Study-specific GWAS analyses stratified by DM status.** We distributed an analysis plan to the 72 participating studies of the CKDGen consortium. Each study conducted analyses separately for individuals with DM and individuals without DM. All studies imputed genotypes to the Haplotype Reference Consortium v1.1 (HRC) or 1000 Genomes Project phase 3 v5 (1000Gp3v5) ALL or phase 1 v3 (1000Gp1v3) ALL panel. Each study conducted linear regression GWAS on log(eGFR) using natural logarithm, an additive genotype model as well as adjusted for sex, age, and other study-specific covariates. Details on study-specific genotyping, imputation, and GWAS were described previously for CKDGen[20]. For UKB, we utilized the fastGWA tool[65] to conduct GWAS for log(eGFR) based on linear mixed models while accounting for sex, age, age × sex, age², age² × sex, and 20 genetic principal components and assuming an additive genetic model, which allowed to include related individuals in the GWAS[65]. For stage 2 studies, MVP, MGI, and HUNT, the same analysis plan was distributed. Details on genotyping, imputation, and GWAS in UKB as well as for stage 2 studies are shown in Supplementary Data 1. For quality control, we excluded variants with low imputation quality, Info < 0.6, or rare variants with minor allele frequency, MAF < 0.1%. We utilized the software packages GWAtoolbox[66] and EasyQC[67] for the quality control of study-specific GWAS results. We conducted a correction for genomic control lambda of the results stratified by diabetes status.

**DM/noDM-stratified GWAS meta-analyses.** In stage 1 of our analysis, separately in DM ($n_{DM} = 88,829$) and noDM ($n_{noDM} = 620,665$) strata, we conducted fixed-effect inverse-variance weighted meta-analyses of 72 GWAS of log(eGFR) using metal[68], and then meta-analyzed these results with DM/noDM-stratified GWAS from UKB ($n_{DM} = 21,040$; $n_{noDM} = 414,628$; European only). To adjust for population stratification within studies, we applied genomic control (GC) correction[69] to each study prior to the meta-analysis. We applied a second GC correction to the DM- and noDM stage 1 meta-analysis results (GC lambda = 1.02 and 1.20, respectively). We excluded variants that were present only in ≤36 stage 1 studies (≤50%) and variants with a cumulative minor allele count of <400 in the stage 1 meta-analyses. In summary, 109,869 individuals with DM and 1,035,190 with noDM were included in stage 1. We followed variants identified at stage 1 in independent stage 2 meta-analyses. For stage 2, we included DM/noDM-stratified GWAS on log(eGFR) from MVP ($n_{DM} = 57,430$, $n_{noDM} = 122,966$, hospital-based), MGI ($n_{DM} = 7469$, $n_{noDM} = 36,558$, hospital-based) and HUNT ($n_{DM} = 3799$, $n_{noDM} = 65,590$, population-based), totalling 68,698 individuals with DM and 225,114 with noDM, all of European-ancestry. Again, we applied study-specific GC correction prior to the meta-analysis and a second GC correction[69] to the stage 2 meta-analysis results (GC lambdas = 1.00 and 1.02 for DM- and noDM, respectively). To maximize power for locus identification, we combined the double GC-corrected stage 1 and 2 meta-analysis separately by DM status via fixed-effect inverse-variance weighted meta-analyses of the two sources using metal[68]. The GC lambda in this final meta-analysis was comparable to previous GWAS[16,70] (GC lambdas = 1.03 and 1.15, respectively). The DM-/noDM-specific summary statistics (stratified GWAS) for each variant genome-wide served to investigate potential DM-/noDM-differential genetic effects on log(eGFR) without making any assumptions on the DM association with any other covariate[71]. These DM-/noDM-specific summary statistics on genetic variants associated with log(eGFR) enabled the implementation of all the following statistical tests to search for DM/noDM-difference loci or novel loci allowing for difference as described below.

**Approaches to search for DM/noDM-difference in genetic effects on eGFR.** We used the meta-analyzed SNP-specific summary statistics, to test for difference in eGFR-effects between DM and noDM. For this, we applied a difference test for each variant using

$$Z_{diff} = \frac{\hat{\beta}_{DM} - \hat{\beta}_{noDM}}{\sqrt{se_{DM}^2 + se_{noDM}^2 - 2r_{diab}se_{DM}se_{noDM}}} \quad (1)$$

where $\hat{\beta}_{DM}$ and $\hat{\beta}_{noDM}$ are the genetic effect estimates for eGFR in DM or noDM from the stratified GWAS meta-analysis, respectively, and corresponding standard errors $se_{DM}$ and $se_{noDM}$. The term $r_{diab}$ reflects the correlation between $\hat{\beta}_{DM}$ and $\hat{\beta}_{noDM}$ across all variants (Spearman correlation coefficient, $r_{diab} = 0.14$).

To search genome-wide for eGFR loci with DM-/noDM-differential effects, we applied two approaches[24,37]: (i) a genome-wide difference test ($P_{Diff} < 5 \times 10^{-8}$, difference test approach), and (ii) a search for genome-wide significant association with overall eGFR followed by a difference test in the same data ($P_{Overall} < 5 \times 10^{-8}$ and $P_{Diff} < 0.05/k$, k = number of followed SNPs, overall+difference test approach). The two approaches complement each other in terms of power to detect difference loci that depend on the magnitude and DM-/noDM specificity[24,37]. We implemented the approaches in two designs: first, we applied a discovery +replication design, where we searched for significant differences in the stage 1 meta-analysis and moved the selected SNPs to a replication stage using the stage 2

meta-analysis (applying a Bonferroni-corrected alpha-level accounting for the SNPs tested). Second, to make full use of stage 1 and stage 2 data, we searched for differences in stage 1 + 2 meta-analysis combined (combined stage design). We utilized the R package EasyStrata[72] to apply the difference test approaches to the DM/noDM-stratified meta-analysis results.

**Approaches to search for eGFR loci allowing for DM/noDM-differences.** By allowing for differences between DM and noDM in SNP-effects on eGFR, one can possibly detect novel loci that have been masked in analyses on overall eGFR (without DM-status stratification). Again, we used the meta-analyzed SNP-specific summary statistics to apply two approaches to search for novel eGFR loci allowing for difference: (i) a screen using a two degrees of freedom joint test ($P_{joint} < 5 \times 10^{-8}$, joint-test approach)[38] derived from a χ²-test using[38]

$$C_{joint} = \left(\frac{\hat{\beta}_{DM}}{se_{DM}}\right)^2 + \left(\frac{\hat{\beta}_{noDM}}{se_{noDM}}\right)^2 \quad (2)$$

where $\hat{\beta}_{DM}$ and $\hat{\beta}_{noDM}$ are the genetic effect estimates for eGFR in DM or noDM from the stratified GWAS meta-analysis, respectively, and corresponding standard errors $se_{DM}$ and $se_{noDM}$; and (ii) a on eGFR association in individuals with or without DM separately (stratified tests approach, $P_{DM} < 5 \times 10^{-8}$ or $P_{noDM} < 5 \times 10^{-8}$). Analogously to the search for difference loci, we applied two-stage designs, the discovery+replication, and the combined stage designs. We utilized the R package EasyStrata[72] to apply the joint and stratified test approaches to the DM/noDM-stratified meta-analysis results.

**Variant selection and region definition.** In order to derive non-overlapping locus regions and locus lead variants, we clumped genome-wide significant variants from the respective test results ($P_{Diff} < 5 \times 10^{-8}$, $P_{Joint} < 5 \times 10^{-8}$, $P_{DM} < 5 \times 10^{-8}$, or $P_{noDM} < 5 \times 10^{-8}$) as done previously[16]: the most significant variant was selected genome-wide (first locus lead variant) and the corresponding locus was defined as the smallest physical interval on the corresponding chromosome containing this variant such that there were no genome-wide significant variants within 500 kb outside the two borders. Omitting the identified locus, we repeated the procedure until no further genome-wide significant variants were detected. By this, a locus region is defined by adding ±250 kb to the first and last genome-wide significant variant of an identified locus. This procedure also ensured non-overlapping loci. A locus was considered to be known for eGFR from previous GWAS, if it overlapped with one of the 427 known eGFR loci (424 from ref.[16], 3 additional from ref.[20]). If a locus is not known, we call it a novel eGFR locus. For the identified variants, we assessed between-study heterogeneity based on the CKDGen meta-analysis using a Chi-Squared test and an I² statistic[73] and verified the association statistics with regards to abnormal or unusually large effect sizes.

**GCTA analyses to identify independent secondary signals within loci.** To evaluate whether there were multiple independent signals within locus, we conducted approximate conditional analyses with GCTA[74] for each identified locus. These analyses were based on European-ancestry individuals, since appropriate trans-ethnic linkage disequilibrium (LD) reference panels were limited. As LD reference panel, we used a random subset of 20,000 unrelated individuals of European-ancestry from UKB, as done previously[16]. To obtain independent signals for associations derived by the difference test, the joint test, or DM- or noDM-specific association tests, we applied a stepwise approach: we conditioned via GCTA analysis on the, respectively, observed lead variant in DM and noDM separately and, second, derived DM/noDM-specific conditioned results for the locus. Separately for DM and noDM, we ensured whether any of the conditioned DM- or noDM-specific associations in the locus showed genome-wide significant association at $P_{DM\_Cond} < 5 \times 10^{-8}$ or $P_{noDM\_Cond} < 5 \times 10^{-8}$. For loci derived by the difference test or the joint test, we applied the difference test or joint test, respectively, to the conditioned DM/noDM estimates to infer whether additional signals showed significant differences or joint effects ($P_{Diff\_Cond} < 5 \times 10^{-8}$ or $P_{Joint\_Cond} < 5 \times 10^{-8}$).

**Prioritization of variants and genes.** For each variant within each identified signal, we derived the variants that were the most likely to drive the association. For this, we calculated approximate Bayes factor and posterior probabilities of association (PPA) based on Z-scores using the Kichaev method[75]. We calculated PPAs based on unconditioned or conditioned summary statistics depending on whether the locus showed only one or multiple independent signals, respectively. Then, we obtained 99% credible sets of variants by sorting the variants within each signal by descending PPA and then summing up PPAs until a cumulative PPA of 99% was reached. To prioritize genes, we used the results from the Gene PrioritiSation (GPS) published previously[16] for known eGFR loci and generated the GPS de novo for novel eGFR loci accordingly: we queried each gene underneath identified loci and prioritized the gene when it mapped to a 99% credible set variant that was protein-relevant with high predicted deleteriousness (CADD[45] PHRED-Score ≥ 15) or expression-relevant in kidney tissue (eQTL, NEPTUNE[46], or GTEx v7[76], false-discovery rate, FDR < 5% for the eQTL) and for splice quantitative trait loci in kidney tissue (sQTL)[76] (FDR < 5% for the sQTL). We also prioritized genes that

were known as human kidney disease monogene: for this, we queried each gene at identified loci for a documented kidney phenotype in human as done previously[16] (Online Mendelian Inheritance in Man® database, OMIM[77] or Groopman et al.[78]) with additional manual curation by expert review to focus on kidney disease monogenes.

**GRS analyses**. GRS analyses based on the previously identified 634 eGFR variants[16] and based on the variants identified for DM-/noDM-difference on eGFR were conducted in unrelated European-ancestry individuals from HUNT ($n_{DM} = 3023$, $n_{noDM} = 25,484$, stage 2 study). For each individual, the GRS was computed as the weighted sum of eGFR-lowering alleles across the 634 variants applying three different weighting schemes: (i) weighted by the respective overall per-variant effect as derived previously (i.e., estimated in CKDGen and UK Biobank, our stage 1 data[16]) (ii) weighted by the variant's DM-/noDM-specific effects as derived here in stage 1 depending on whether the individual in HUNT had DM or noDM, respectively (and vice versa), (iii) weighted by the variant's DM-/noDM-specific effect from stage 1 for HUNT individuals with DM/noDM, respectively, when the variant was among the seven with an identified significant difference, and weighted by the overall-effect size otherwise. By this, the GRS association analyses conducted in a stage 2 study were independent of the variant identification and weight estimation, which were based on stage 1 studies. Separately for individuals with and without DM, we estimated the association of the GRS on eGFR by linear regression (original scale; adjusted by sex, age, and genetic principal components). We judged the GRS effect per standard deviation of the GRS and the eGFR variance explained by the GRS separately in DM and noDM separately.

**Reporting summary**. Further information on research design is available in the Nature Research Reporting Summary linked to this article.

## Data availability
Summary genetic association results for the DM-status-specific meta-analyses for log(eGFRcrea) can be downloaded from https://ckdgen.imbi.uni-freiburg.de/. All other data are available from the corresponding author on reasonable request.

## Code availability
The analysis plan can be downloaded from https://ckdgen.eurac.edu/mediawiki/index.php/CKDGen_Round_4_EPACTS_analysis_plan and the phenotype command line script from https://github.com/genepi-freiburg/ckdgen-pheno. All other code is available from the corresponding author on reasonable request.

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

## Acknowledgements

The Deutsche Forschungsgemeinschaft (DFG, German Research Foundation) supported the meta-analysis—Project-ID 387509280—SFB1350 (Subproject C6 to I.M.H.). A.M.H., B.R., and R.T. were supported by VACSR&D MVP grant CX001897. This research is based on data from the Million Veteran Program, Office of Research and Development, Veterans Health Administration, and was supported by VACSR&D MVP grant CX001897 (A.M.H.). This publication does not represent the views of the Department of Veteran Affairs or the United States Government. We conducted this research using the UK Biobank resource under the application number 20272. We thank Paola Bilani for collecting author information. Extended acknowledgements are provided in Supplementary Note 4 for all studies, in Supplementary Note 5 for MVP and in Supplementary Note 6 for LifeLines.

## Author contributions

T.W., H.R., A.Teu., M.Go., B.R., K.Stan., R.T., U.Th., C.A.B., A.Hu., F.K., A.Köt., C.P., and I.M.H. wrote the manuscript. T.W., H.R., A.Teu., M.Go., C.A.B., A.Köt., C.P., and I.M.H. designed the study. B.T., D.Cu., K.B.S., M.Sch., Q.Y., A.B.Z., A.Ca., A.Co., A.d.G., A.Fer., A.J.O., A.Kör., A.Ma., A.Me., A.P.J.d.V., A.Pa., A.Pe., A.R.R., A.Tö., B.B., B.Br., B.I.F., B.J., B.K.K., B.M.P., B.P., B.Sch., B.Sp., B.W.J.H.P., C.G., C.H., C.K., C.M.K., C.N.S., C.R., C.Sa., C.W., Ch.F., D.F.R., D.I.C., D.M., D.T., E.B., E.S.T., F.Gi., G.E., G.G., G.N.N., G.S., G.We., H.B., H.G., H.H., H.Kr., H.L., H.M.S., H.Sc., I.M.N., I.R., I.T., J.Ä., J.B.J., J.C.C., J.Co., J.F.F., J.F.W., J.G., J.Ha., J.J.M.L., J.O., J.P.C., J.S.K., J.Th., J.X., K.Ch., K.D., K.Ec., K.Ho., K.Hv., K.L.M., K.Ste., K.Str., L.J.L., L.L., L.M.Y.A., L.Wal., M.A.I., M.A.L., M.A.P., M.Boc., M.Ci., M.E.K., M.Gö., M.H.d.B., M.K.E., M.Käh., M.Kub., M.Lo., M.N., M.O.M., M.Per., M.Y., N.G.M., N.Pi., N.S.J., N.V., O.D., O.M., O.P., O.T.R., P.E., P.G., P.H., P.K., P.K.E.M., P.M.R., P.Ma., P.O.S., P.P.P., P.R., P.Vo., R.d.M., R.Do., R.J.C., R.J.F.L., R.S., S.C.L., S.F.W., S.Ha., S.J.L.B., S.L., S.Va., T.B., T.E., T.L., T.S.A., T.Wo., U.V., V.Gi., V.Gu., V.H.X.F., V.S., V.W.V.J., v.D.R.M., W.B.W., W.Ki., W.Ko., W.L., W.Mä., W.Zh., X.S., Y.Fri., Y.M., Y.X.W., K.Star., U.Th., C.A.B., A.Hu., F.K., A.Köt., and C.P. managed an individual contributing study. T.W., H.R., A.Teu., M.Go., B.R., K.Stan., L.T., A.Ch., A.Ti., B.T., C.H.L.T., J.B.W., J.F.C., K.Hor., M.Co., M.Li., M.Sch., M.Wu., P.J.v.d.M., Q.Y., S.Gh., T.N., Y.L., F.Gue., A.D., A.P.J.d.V., A.P.M., A.V.S., B.I.F., B.J., B.N., B.V., C.D.L., C.L., C.M.L., C.N.S., C.Sc., C.Y.C., Ch.F., Ch.Wa., D.Cz., D.F.G., D.I.C., D.O.M.K., E.H., E.S., F.Riv., F.Riz., G.D., G.G., G.S., H.D.W., H.K.i., H.Sn., I.M.N., J.Ä., J.B.W., J.D., J.F.F., J.J., J.Ku., J.L., J.Marte., J.Marti., J.Tr., K.A.R., K.M.R., K.Mi., L.A.L., L.L., L.Ly., L.M.R., L.Wan., M.A., M.A.H., M.Boc., M.E.K., M.F., M.F.F., M.Ga., M.H.P., M.K.W., M.Kan., M.Kuo., M.L.Big., M.L.O., M.P.C., M.Stu., M.LiC., N.M., P.H., P.K.J., P.P.M., R.J.C., R.Ru., R.T., R.T.G., R.Z.H.S., S.A.P., S.C.F., S.D.G., S.Gr., S.Pat., S.Se., S.Va., S.Vo., Sh.J.H., T.B., T.C., T.H., T.S.A., Th.SB., V.V., W.Mä., W.Zh., X.S., Y.Ch.T., Y.K., Y.M., Y.O., K.Star., C.A.B., A.Köt., C.P., and I.M.H. performed statistical methods and analysis. T.W., H.R., M.Go., B.R., K.Stan., L.T., A.Ch., A.Ho., J.B.W., J.F.C., K.Hor., M.Co., M.Li., M.Sch., M.Wu., P.J.v.d.M., S.Gh., T.N., Y.L., G.P., A.D., A.I.P., A.P.M., A.V.S., B.V., C.A.H., C.D.L., C.N.S., C.Sc., C.Y.C., Ch.Wa., D.Cz., E.C., E.H., E.S., E.W.A., F.Riz., G.E., G.Pi., H.C., H.G., H.Ki., J.B.W., J.D., J.J., J.L., J.Marte., J.Marti., J.Tr., K.L.M., L.C., L.Ly., M.E.K., M.F., M.H.P., M.P.C., M.Stu., M.LiC., M.Ling.C., N.S.J., P.H., P.P.M., R.M., R.Ru., R.T., R.T.G., S.Ber., S.Gr., S.Pat., S.Se., T.C., T.S.A., V.H.X.F., W.Mä., Zh.Y., C.A.B., and I.M.H. performed bioinformatics. T.W., H.R., A.Teu., M.Go., B.R., K.Stan., J.F.C., K.Hor., M.Sch., M.Wu., S.Gh., Y.L., A.Ma., A.Tö., B.A., B.I.F., B.J., B.Sch., C.D.L., C.G., C.L., C.M.L., C.N.S., C.Y.C., D.F.R., D.I.C., F.Guo., H.B., H.H., H.Ki., I.T., J.Ä., J.B.W., J.D., J.F.F., J.Ha., J.Marte., J.Tr., K.D., K.En., K.L.M., K.Mi., L.L., M.F.F., M.Ga., M.Stu., N.D.P., N.V., N.Y.Q.T., P.H., P.v.d.H., R.T., R.T.G., R.Z.H.S., S.Se., S.Va., S.Vo., Sh.J.H., T.B., T.S.A., V.V., W.H., W.Ko., X.G., K.Star., U.Th., C.A.B., A.Hu., A.Köt., C.P., and I.M.H. interpreted results. A.Teu., A.A.H., A.B.Z., A.Ca., A.Co., A.d.G., A.Fra., A.Kör., A.Mo., B.H.S., B.W.J.H.P., C.H., C.N.S., C.W., Ch.Wa., D.F.R., D.I.C., D.J.P.,

D.T., E.B., E.S., E.W.A., F.Gi., F.Guo., F.Riv., F.Riz., G.E., H.B., H.C., H.G., J.Ä., J.B.W., J.C.C., J.C.M., J.I.R., J.P.C., J.S.K., J.Tr., K.D.T., K.Hv., K.L.M., L.A.L., L.C., L.L., L.Ly., L.Wan., M.L.B., M.A.N., M.Boe., M.E.K., M.F.F., M.H.P., M.K.E., M.K.W., M.Käh., M.Kub., M.La., M.N., M.O.M., M.Per., M.Wa., M.Ling.C., N.D.P., N.M., N.Pi., N.S.J., N.V., O.M., O.T.R., P.H., P.K.E.M., P.M.R., P.O.S., P.S., P.v.d.H., R.B., R.J.C., R.N., R.Z.H.S., S.Pad., S.Va., T.J.R., T.L., T.M., U.V., V.H.X.F., V.S., v.D.R.M., W.H., W.Ko., W.Mä., Y.Fri., Y.X.W., K.Star., C.A.B., A.Hu., and F.K. performed genotyping. D.Cu., A.B.Z., A.Ca., A.Co., A.d.G., A.Ma., A.Me., A.R.R., A.Ter., A.Tö., B.H.S., B.I.F., B.J., B.P., B.W.J.H.P., C.L., C.M.K., C.M.S., C.Sa., Ch.K.H., D.J.P., D.M., D.M.W., D.R., E.P.B., E.S.T., G.B., G.W.M., G.We., H.B., H.H., I.O., I.R., I.T., J.B.J., J.C.C., J.Co., J.Ha., J.J.M.L., J.P.C., J.S.K., K.Ch., K.D., K.Ec., K.Ho., K.Hv., K.Ma., K.N., L.J.L., L.K., M.A.P., M.Ci., M.Gö., M.I., M.K.E., M.Käh., M.Kas., M.Kub., M.Per., N.H.K., N.Pi., O.P., O.T.R., P.H., P.M.R., P.Ma., P.O.S., P.P.P., P.R., P.S., P.Vo., R.S., S.C.L., S.F.W., S.Pad., S.Vo., T.B., T.J.R., T.L., T.Wo., V.Gu., V.S., V.W.V.J., v.D.R.M., W.H., W.Mä., K.Star., U.Th., C.A.B., A.Köt., and C.P. recruited subjecs. T.W., H.R., A.Teu., M.Go., B.R., K.Stan., L.T., A.Ch., A.Ho., A.Ti., B.T., C.H.L.T., D.Cu., J.B.W., J.F.C., K.B.S., K.Hor., M.Co., M.Li., M.Sch., M.Wu., P.J.v.d.M., Q.Y., S.Gh., T.N., Y.L., G.P., F.Gue., A.A.H., A.B.Z., A.Ca., A.Co., A.D., A.d.G., A.Fer., A.Fra., A.I.P., A.J.O., A.Kör., A.Ma., A.Me., A.Mo., A.P.J.d.V., A.P.M., A.Pa., A.Pe., A.R.R., A.Ter., A.Tö., A.V.S., B.A., B.B., B.Br., B.H.S., B.I.F., B.J., B.K.K., B.M.P., B.N., B.P., B.Sch., B.Sp., B.V., B.W.J.H.P., C.A.H., C.D.L., C.G., C.H., C.K., C.L., C.M.K., C.M.L., C.M.S., C.N.S., C.R., C.Sa., C.Sc., C.W., C.Y.C., Ch.F., Ch.K.H., Ch.Wa., D.Cz., D.F.G., D.F.R., D.I.C., D.J.P., D.M., D.M.W., D.O.M.K., D.R., D.T., E.B., E.C., E.H., E.P.B., E.S., E.S.T., E.W.A., F.Gi., F.Guo., F.Riv., F.Riz., G.B., G.D., G.E., G.G., G.N.N., G.Pi., G.S., G.W.M., G.We., H.B., H.C., H.D.W., H.G., H.H., H.Ki., H.Kr., H.L., H.M.S., H.Sc., H.Sn., I.M.N., I.O., I.R., I.T., J.Ä., J.B.J., J.B.W., J.C.C., J.C.M., J.Co., J.D., J.F.F., J.F.W., J.G., J.Ha., J.I.R., J.J., J.J.M.L., J.Ku., J.L., J.Marte., J.Marti., J.O., J.P.C., J.S.K., J.Th., J.Tr., J.X., K.A.R., K.Ch., K.D., K.D.T., K.Ec., K.En., K.Ho., K.Hv., K.L.M., K.M.R., K.Ma., K.Mi., K.N., K.Ste., K.Str., L.A.L., L.C., L.J.L., L.K., L.L., L.L.y., L.M.R., L.M.Y.A., L.P., L.Wal., L.Wan., M.L.B., M.A., M.A.I., M.A.L., M.A.N., M.A.P., M.Boc., M.Boe., M.Ci., M.E.K., M.F., M.F.F., M.Ga., M.Gö., M.H.d.B., M.H.P., M.I., M.K., M.K.E., M.K.W., M.Käh., M.Kan., M.Kas., M.Kub., M.Kuo., M.L.Big., M.L.O., M.La., M.Lo., M.N., M.O.M., M.P.C., M.Per., M.Pir., M.Stu., M.Wa., M.Y., M.LiC., M.Ling.C., N.B., N.D.P., N.G.M., N.H.K., N.M., N.Pi., N.S.J., N.Sch., N.V., N.Y.Q.T., O.D., O.M., O.P., O.T.R., P.E., P.G, P.H., P.K, P.K.E.M., P.K.J., P.M.R., P.Ma., P.O.S., P.P.M., P.P.P., P.R., P.S., P.v.d.H., P.Vo., R.B., R.d.M., R.Do., R.J.C., R.J.F.L., R.M., R.N., R.Ru., R.S., R.T., R.T.G., R.Z.H.S., S.A.P., S.Ber., S.C.F., S.C.L., S.D.G., S.F.W., S.Gr., S.Ha., S.J.L.B., S.L., S.Pad., S.Pat., S.Se., S.Va., S.Vo., Sh.J.H., Sh.M.K., T.B., T.C., T.E., T.H., T.J.R., T.L., T.M., T.S.A., T.Wo., Th.S.B., U.V., V.Gi., V.Gu., V.H.X.F., V.S., V.V., V.W.V.J., v.D.R.M., W.B.W., W.H., W.Ki., W.Ko., W.L., W.Mä., W.Zh., X.G., X.S., Y.Ch.T., Y.Fri., Y.K., Y.M., Y.O., Y.X.W., Zh.Y., K.Star., U.Th., C.A.B., A.Hu., F.K., A.Köt., C.P., and I.M.H. critically reviewed the manuscript.

## Funding

## Competing interests

GlaxoSmithKline and Merck & Co employed A.Y.C. Janssen Pharmaceuticals and GlaxoSmithKline employed D.M.W. K.B.S., L.M.Y.-A. and M.A.L. are full-time employees of GlaxoSmithKline. M.S. receives funding from Pfizer Inc. for a project not related to this research. J.Ä. reports personal fees from AstraZeneca, Boehringer Ingelheim and Novartis, outside of the submitted work. D.F.G., H.H., K.S., P.S., G.S. and U.T. are employees of deCODE/Amgen Inc. Kevin Ho received support by Fresenius Medical Care North America. M.K. is employed with Synlab Holding Deutschland GmbH. W.K. reports consulting fees from AstraZeneca, Novartis, Pfizer, The Medicines Company, DalCor, Kowa, Amgen, Corvidia, Daiichi-Sankyo, Genentech, Novo Nordisk, Esperion, OMEICOS, LIB Therapeutics, speaker honoraria from Amgen, AstraZeneca, Novartis, Berlin-Chemie, Sanofi, and Bristol-Myers Squibb, and grants and non-financial support from Abbott, Roche Diagnostics, Beckmann, and Singulex, outside the submitted work. C.L. received Grants/ Research Support from Bayer Ag/ Novo Nordisk, Husband works for Vertex. As of January 2020, A.M. is an employee of Genentech, and a holder of Roche stock. W.M. is employed with Synlab Holding Deutschland GmbH. D.O.M.-K. is a partime research physician at Metabolon, Inc. M.A.N. was supported by a consulting contract between Data Tecnica International LLC and the National Institute on Aging (NIA), National Institutes of Health (NIH), Bethesda, MD, USA and consults for a number of small biotech and pharma. M.L.O. received grant support from GlaxoSmithKline during conduct of the study and received support from Novartis, Merck, Amgen, and AstraZeneca. L.S.P. has served on Scientific Advisory Boards for Janssen, and has or had research support from Merck, Pfizer, Eli Lilly, Novo Nordisk, Sanofi, PhaseBio, Roche, Abbvie, Vascular Pharmaceuticals, Janssen, Glaxo SmithKline, and the Cystic Fibrosis Foundation. He is also a cofounder, Officer and Board member and stockholder for a company, Diasyst, Inc., which markets software aimed to help improve diabetes management. A.I.P. and D.F.R. are employees of Merck Sharp Dohme Corp. Bruce.M.P. serves on the steering committee of the Yale Open Data Access Project funded by Johnson & Johnson. P.R. received fees to his institution for research support from AstraZeneca and Novo Nordisk; for steering group participation from AstraZeneca, Gilead, Novo Nordisk, and Bayer; for lectures from Bayer, Eli Lilly and Novo Nordisk; and for advisory boards from Sanofi and Boehringer Ingelheim outside of this work. V.S. has received a modest honorarium from Sanofi for consulting. He also has ongoing research collaboration with Bayer Ltd. (all outside of the present study). L.W. received institutional grants from GlaxoSmithKline, AstraZeneca, BMS, Boehringer-Ingelheim, Pfizer, MSD and Roche Diagnostics. H.W. has received grant support paid to the institution and fees for serving on Steering Committees of the ODYSSEY trial from Sanofi and Regeneron Pharmaceuticals, the ISCHEMIA and the MINT studies from the National Institutes of Health, the STRENGTH trial from Omthera Pharmaceuticals, the HEART-FID study from American Regent, the DAL-GENE study from DalCor Pharma UK Inc., the AEGIS-II study from CSL Behring, the SCORED and SOLOIST-WHF from Sanofi Aventis Australia Pty. Ltd., and the CLEAR OUTCOMES study from Esperion Therapeutics. M.P. is partly funded by the study FinnGen (www.finngen.fi), which is jointly funded by a Finnish Governmental agency Business Finland and thirteen international pharmaceutical companies: Abbvie, AstraZeneca, Biogen, Boehringer Ingelheim, Bristol-Myers Squibb, Genentech, a member of the Roche Group, GlaxoSmithKline (GSK), Janssen, Maze Therapeutics, MSD (the tradename of Merck & Co., Inc, Kenilworth, NJ USA), Novartis, Pfizer and Sanofi. C.C.K. is an Editorial Board Member for Communications Biology, but was not involved in the editorial review of, nor the decision to publish this article. The remaining authors declare no competing interests.

## Additional information

Thomas W. Winkler [1,266]✉, Humaira Rasheed [2,3,4,266], Alexander Teumer [5,6,7,266], Mathias Gorski [1,8,266], Bryce X. Rowan [9,10,266], Kira J. Stanzick [1], Laurent F. Thomas [2,11,12], Adrienne Tin [13,14], Anselm Hoppmann [15], Audrey Y. Chu [16], Bamidele Tayo [17], Chris H. L. Thio [18], Daniele Cusi [19,20], Jin-Fang Chai [21], Karsten B. Sieber [22],

Katrin Horn[23,24], Man Li[25], Markus Scholz[23,24], Massimiliano Cocca[26], Matthias Wuttke[15,27], Peter J. van der Most[18], Qiong Yang[28], Sahar Ghasemi[5,6,29], Teresa Nutile[30], Yong Li[15], Giulia Pontali[31,32], Felix Günther[1,33], Abbas Dehghan[34,35], Adolfo Correa[13], Afshin Parsa[36,37], Agnese Feresin[38], Aiko P. J. de Vries[39], Alan B. Zonderman[40], Albert V. Smith[41,42], Albertine J. Oldehinkel[43], Alessandro De Grandi[31], Alexander R. Rosenkranz[44], Andre Franke[45], Andrej Teren[24,46], Andres Metspalu[47], Andrew A. Hicks[31], Andrew P. Morris[48,49,50], Anke Tönjes[51], Anna Morgan[26], Anna I. Podgornaia[16], Annette Peters[52,53,54], Antje Körner[24,55,56], Anubha Mahajan[49], Archie Campbell[57], Barry I. Freedman[58], Beatrice Spedicati[38], Belen Ponte[59], Ben Schöttker[60,61], Ben Brumpton[2,62,63], Bernhard Banas[8], Bernhard K. Krämer[64], Bettina Jung[7,8,65], Bjørn Olav Åsvold[2,66], Blair H. Smith[67], Boting Ning[28], Brenda W. J. H. Penninx[68], Brett R. Vanderwerff[69,70], Bruce M. Psaty[71,72,73], Candace M. Kammerer[74], Carl D. Langefeld[75], Caroline Hayward[76], Cassandra N. Spracklen[77,78], Cassianne Robinson-Cohen[10,79], Catharina A. Hartman[43], Cecilia M. Lindgren[80,81,82], Chaolong Wang[83,84], Charumathi Sabanayagam[85,86], Chew-Kiat Heng[87,88], Chiara Lanzani[89], Chiea-Chuen Khor[83,85], Ching-Yu Cheng[85,86,90], Christian Fuchsberger[31], Christian Gieger[52,53,91], Christian M. Shaffer[92], Christina-Alexandra Schulz[93], Cristen J. Willer[94,95,96], Daniel I. Chasman[97,98], Daniel F. Gudbjartsson[99,100], Daniela Ruggiero[30,101], Daniela Toniolo[102], Darina Czamara[103], David J. Porteous[57,104], Dawn M. Waterworth[22], Deborah Mascalzoni[31,105], Dennis O. Mook-Kanamori[106,107], Dermot F. Reilly[16], E. Warwick Daw[108], Edith Hofer[109,110], Eric Boerwinkle[111], Erika Salvi[112], Erwin P. Bottinger[113,114], E-Shyong Tai[21,115,116], Eulalia Catamo[26], Federica Rizzi[20,117], Feng Guo[60], Fernando Rivadeneira[118,119], Franco Guilianini[97], Gardar Sveinbjornsson[99], Georg Ehret[120], Gerard Waeber[121], Ginevra Biino[122], Giorgia Girotto[26,38], Giorgio Pistis[123], Girish N. Nadkarni[113,124], Graciela E. Delgado[125], Grant W. Montgomery[126], Harold Snieder[18], Harry Campbell[127], Harvey D. White[128], He Gao[34], Heather M. Stringham[129], Helena Schmidt[130], Hengtong Li[85], Hermann Brenner[60,61], Hilma Holm[99], Holgen Kirsten[23,24], Holly Kramer[17,131], Igor Rudan[127], Ilja M. Nolte[18], Ioanna Tzoulaki[34,35,132], Isleifur Olafsson[133], Jade Martins[103], James P. Cook[48], James F. Wilson[76,127], Jan Halbritter[51,134], Janine F. Felix[119,135], Jasmin Divers[75], Jaspal S. Kooner[136,137,138,139], Jeannette Jen-Mai Lee[21], Jeffrey O'Connell[37], Jerome I. Rotter[140], Jianjun Liu[83,115], Jie Xu[141], Joachim Thiery[24,142], Johan Ärnlöv[143,144], Johanna Kuusisto[145,146], Johanna Jakobsdottir[147,148], Johanne Tremblay[149,150], John C. Chambers[34,136,137,138,151], John B. Whitfield[152], John M. Gaziano[153,154], Jonathan Marten[76], Josef Coresh[14], Jost B. Jonas[141,155,156,157], Josyf C. Mychaleckyj[158], Kaare Christensen[159], Kai-Uwe Eckardt[160,161], Karen L. Mohlke[77], Karlhans Endlich[6,162], Katalin Dittrich[55,56], Kathleen A. Ryan[163], Kenneth M. Rice[164], Kent D. Taylor[140], Kevin Ho[165,166], Kjell Nikus[167,168], Koichi Matsuda[169], Konstantin Strauch[170,171,172], Kozeta Miliku[119,135], Kristian Hveem[2], Lars Lind[173], Lars Wallentin[174,175], Laura M. Yerges-Armstrong[22], Laura M. Raffield[77], Lawrence S. Phillips[176,177], Lenore J. Launer[178], Leo-Pekka Lyytikäinen[179,180], Leslie A. Lange[181], Lorena Citterio[89], Lucija Klaric[76], M. Arfan Ikram[182], Marcus Ising[183], Marcus E. Kleber[125,184], Margherita Francescatto[38], Maria Pina Concas[26], Marina Ciullo[30,101], Mario Piratsu[185], Marju Orho-Melander[93], Markku Laakso[145,146], Markus Loeffler[23,24], Markus Perola[186,187], Martin H. de Borst[188], Martin Gögele[31], Martina La Bianca[26], Mary Ann Lukas[189], Mary F. Feitosa[108], Mary L. Biggs[71,164], Mary K. Wojczynski[108], Maryam Kavousi[182], Masahiro Kanai[190,191], Masato Akiyama[190,192], Masayuki Yasuda[85,193], Matthias Nauck[6,194], Melanie Waldenberger[52,91,195], Miao-Li Chee[85], Miao-Ling Chee[85], Michael Boehnke[129], Michael H. Preuss[113], Michael Stumvoll[51], Michael A. Province[108], Michele K. Evans[40], Michelle L. O'Donoghue[196,197], Michiaki Kubo[198], Mika Kähönen[199,200], Mika Kastarinen[146], Mike A. Nalls[201,202], Mikko Kuokkanen[187,203,204], Mohsen Ghanbari[182,205],

Murielle Bochud[206], Navya Shilpa Josyula[207], Nicholas G. Martin[152], Nicholas Y. Q. Tan[85], Nicholette D. Palmer[208], Nicola Pirastu[127], Nicole Schupf[209], Niek Verweij[210], Nina Hutri-Kähönen[211], Nina Mononen[179,180], Nisha Bansal[212,213], Olivier Devuyst[214], Olle Melander[93], Olli T. Raitakari[215,216,217], Ozren Polasek[218,219], Paolo Manunta[89], Paolo Gasparini[26,38], Pashupati P. Mishra[179,180], Patrick Sulem[99], Patrik K. E. Magnusson[220], Paul Elliott[34,35,221,222], Paul M. Ridker[97,98], Pavel Hamet[149,223], Per O. Svensson[224,225], Peter K. Joshi[127], Peter Kovacs[51,226], Peter P. Pramstaller[31], Peter Rossing[227,228], Peter Vollenweider[121], Pim van der Harst[210,229], Rajkumar Dorajoo[83], Ralene Z. H. Sim[85], Ralph Burkhardt[24,142,230], Ran Tao[9,231], Raymond Noordam[232], Reedik Mägi[47], Reinhold Schmidt[109], Renée de Mutsert[107], Rico Rueedi[233,234], Rob M. van Dam[21,235], Robert J. Carroll[92], Ron T. Gansevoort[188], Ruth J. F. Loos[113,236,237], Sala Cinzia Felicita[102], Sanaz Sedaghat[182], Sandosh Padmanabhan[238], Sandra Freitag-Wolf[239], Sarah A. Pendergrass[240], Sarah E. Graham[94], Scott D. Gordon[152], Shih-Jen Hwang[241,242], Shona M. Kerr[76], Simona Vaccargiu[185], Snehal B. Patil[69,70,95], Stein Hallan[11,243], Stephan J. L. Bakker[188], Su-Chi Lim[21,244], Susanne Lucae[183], Suzanne Vogelezang[119,135], Sven Bergmann[233,234], Tanguy Corre[206,233,234], Tarunveer S. Ahluwalia[227,245], Terho Lehtimäki[179,180], Thibaud S. Boutin[76], Thomas Meitinger[195,246,247], Tien-Yin Wong[85,86,90], Tobias Bergler[8], Ton J. Rabelink[39,248], Tõnu Esko[47,249], Toomas Haller[47], Unnur Thorsteinsdottir[41,99], Uwe Völker[6,250], Valencia Hui Xian Foo[85], Veikko Salomaa[186], Veronique Vitart[76], Vilmantas Giedraitis[251], Vilmundur Gudnason[41,147], Vincent W. V. Jaddoe[119,135], Wei Huang[252,253], Weihua Zhang[34,136], Wen Bin Wei[254], Wieland Kiess[24,55,56], Winfried März[125,255,256], Wolfgang Koenig[195,257,258], Wolfgang Lieb[259], Xin Gao[60], Xueling Sim[21], Ya Xing Wang[141], Yechiel Friedlander[260], Yih-Chung Tham[85], Yoichiro Kamatani[190,261], Yukinori Okada[190,262,263], Yuri Milaneschi[68], Zhi Yu[14,81,264], Lifelines cohort study*, DiscovEHR/MyCode study*, VA Million Veteran Program*, Klaus J. Stark[1], Kari Stefansson[41,99], Carsten A. Böger[7,8,65], Adriana M. Hung[10,79,267], Florian Kronenberg[265,267], Anna Köttgen[14,15,267], Cristian Pattaro[31,267] & Iris M. Heid[1,267 ✉]

[1]Department of Genetic Epidemiology, University of Regensburg, Regensburg, Germany. [2]K. G. Jebsen Center for Genetic Epidemiology, Department of Public Health and Nursing, Faculty of Medicine and Health Sciences, NTNU, Norwegian University of Science and Technology, Trondheim, Norway. [3]MRC Integrative Epidemiology Unit, Population Health Sciences, Bristol Medical School, University of Bristol, Bristol, UK. [4]Division of Medicine and Laboratory Sciences, University of Oslo, Oslo, Norway. [5]Institute for Community Medicine, University Medicine Greifswald, Greifswald, Germany. [6]DZHK (German Center for Cardiovascular Research), partner site Greifswald, Greifswald, Germany. [7]Department of Population Medicine and Lifestyle Diseases Prevention, Medical University of Bialystok, Bialystok, Poland. [8]Department of Nephrology, University Hospital Regensburg, Regensburg, Germany. [9]Department of Biostatistics, Vanderbilt University Medical Center, Nashville, TN, USA. [10]Department of Veteran's Affairs, Tennessee Valley Healthcare System (626)/Vanderbilt University, Nashville, TN, USA. [11]Department of Clinical and Molecular Medicine, NTNU, Norwegian University of Science and Technology, Trondheim, Norway. [12]BioCore—Bioinformatics Core Facility, Norwegian University of Science and Technology, Trondheim, Norway. [13]Department of Medicine, University of Mississippi Medical Center, Jackson, MS, USA. [14]Department of Epidemiology, Johns Hopkins Bloomberg School of Public Health, Baltimore, MD, USA. [15]Institute of Genetic Epidemiology, Department of Data Driven Medicine, Faculty of Medicine and Medical Center–University of Freiburg, Freiburg, Germany. [16]Genetics, Merck & Co., Inc, Kenilworth, NJ, USA. [17]Department of Public Health Sciences, Loyola University Chicago, Maywood, IL, USA. [18]Department of Epidemiology, University of Groningen, University Medical Center Groningen, Groningen, The Netherlands. [19]Institute of Biomedical Technologies, National Research Council of Italy, Milan, Italy. [20]Bio4Dreams—Business Nursery for Life Sciences, Milan, Italy. [21]Saw Swee Hock School of Public Health, National University of Singapore and National University Health System, Singapore, Singapore. [22]Target Sciences—Genetics, GlaxoSmithKline, Collegeville, PA, USA. [23]Institute for Medical Informatics, Statistics and Epidemiology, University of Leipzig, Leipzig, Germany. [24]LIFE Research Center for Civilization Diseases, University of Leipzig, Leipzig, Germany. [25]Division of Nephrology and Hypertension, Department of Medicine, University of Utah, Salt Lake City, UT, USA. [26]Institute for Maternal and Child Health, IRCCS 'Burlo Garofolo', Trieste, Italy. [27]Renal Division, Department of Medicine IV, Faculty of Medicine and Medical Center—University of Freiburg, Freiburg, Germany. [28]Department of Biostatistics, Boston University School of Public Health, Boston, MA, USA. [29]Department of Psychiatry and Psychotherapy, University Medicine Greifswald, Greifswald, Germany. [30]Institute of Genetics and Biophysics 'Adriano Buzzati-Traverso'—CNR, Naples, Italy. [31]Eurac Research, Institute for Biomedicine (affiliated with the University of Lübeck), Bolzano, Italy. [32]University of Trento, Department of Cellular, Computational and Integrative Biology—CIBIO, Trento, Italy. [33]Statistical Consulting Unit StaBLab, Department of Statistics, LMU Munich, Munich, Germany. [34]MRC Centre for Environment and Health, Department of Epidemiology and Biostatistics, School of Public Health, Faculty of Medicine, Imperial College London, London, UK. [35]Dementia Research Institute, Imperial College London, London, UK. [36]Division of Kidney, Urologic and Hematologic Diseases, National Institute of Diabetes and Digestive and Kidney Diseases, National Institutes of Health, Bethesda, MD, USA. [37]University of Maryland School of Medicine, Baltimore, MD, USA. [38]Department of Medicine, Surgery and Health Sciences, University of Trieste, Trieste, Italy. [39]Section of Nephrology, Department of Internal Medicine, Leiden University Medical Center, Leiden, The Netherlands. [40]Laboratory of Epidemiology and Population Sciences, National Institute on Aging, Intramural Research Program, US National Institutes of Health, Baltimore, MD, USA. [41]Faculty of Medicine, School of Health Sciences, University of Iceland, Reykjavik, Iceland. [42]CNRS UMR 8199, European Genomic Institute for

Diabetes (EGID), Institut Pasteur de Lille, University of Lille, Lille, France. [43]Interdisciplinary Center of Psychopathology and Emotion Regulation (ICPE), University of Groningen, University Medical Center Groningen, Groningen, The Netherlands. [44]Department of Internal Medicine, Division of Nephrology, Medical University Graz, Graz, Austria. [45]Institute of Clinical Molecular Biology, Christian-Albrechts-University of Kiel, Kiel, Germany. [46]Heart Center Leipzig, Leipzig, Germany. [47]Estonian Genome Centre, Institute of Genomics, University of Tartu, Tartu, Estonia. [48]Department of Health Data Science, University of Liverpool, Liverpool, UK. [49]Wellcome Centre for Human Genetics, University of Oxford, Oxford OX3 7BN, UK. [50]Centre for Genetics and Genomics Versus Arthritis, Centre for Musculoskeletal Research, The University of Manchester, Manchester, UK. [51]Medical Department III—Endocrinology, Nephrology, Rheumatology, University of Leipzig Medical Center, Leipzig, Germany. [52]Institute of Epidemiology, Helmholtz Zentrum München—German Research Center for Environmental Health, Neuherberg, Germany. [53]German Center for Diabetes Research (DZD), Neuherberg, Germany. [54]Chair of Epidemiology, IBE, Faculty of Medicine, Ludwig-Maximilians-Universität München, München, Germany. [55]Department of Women and Child Health, Hospital for Children and Adolescents, University of Leipzig, Leipzig, Germany. [56]Center for Pediatric Research, University of Leipzig, Leipzig, Germany. [57]Center for Genomic and Experimental Medicine, Institute of Genetics and Cancer, University of Edinburgh, Edinburgh, UK. [58]Section on Nephrology, Internal Medicine, Wake Forest School of Medicine, Winston-Salem, NC, USA. [59]Service de Néphrologie et Hypertension, Medicine Department, Geneva University Hospitals, Geneva, Switzerland. [60]Division of Clinical Epidemiology and Aging Research, German Cancer Research Center (DKFZ), Heidelberg, Germany. [61]Network Aging Research, University of Heidelberg, Heidelberg, Germany. [62]Clinic of Medicine, St. Olavs Hospital, Trondheim University Hospital, Trondheim 7030, Norway. [63]HUNT Research Centre, Department of Public Health and Nursing, NTNU, Norwegian University of Science and Technology, Levanger 7600, Norway. [64]Vth Department of Medicine (Nephrology, Hypertensiology, Endocrinology, Diabetology, Rheumatology, Pneumology), Medical Faculty Mannheim, University of Heidelberg, Mannheim, Germany. [65]Department of Nephrology and Rheumatology, Kliniken Südostbayern, Traunstein, Germany. [66]Department of Endocrinology, Clinic of Medicine, St. Olavs Hospital, Trondheim University Hospital, Trondheim, Norway. [67]Division of Population Health and Genomics, Ninewells Hospital and Medical School, University of Dundee, Dundee, UK. [68]Department of Psychiatry, VU University Medical Centre, Amsterdam, The Netherlands. [69]Department of Biostatistics, University of Michigan School of Public Health, Ann Arbor, MI 48109, USA. [70]Center for Statistical Genetics, University of Michigan School of Public Health, Ann Arbor, MI 48109, USA. [71]Cardiovascular Health Research Unit, Department of Medicine, University of Washington, Seattle, WA, USA. [72]Department of Epidemiology, University of Washington, Seattle, WA, USA. [73]Department of Health Systems and Population Health, University of Washington, Seattle, WA, USA. [74]Department of Human Genetics, Graduate School of Public Health, University of Pittsburgh, Pittsburgh, PA, USA. [75]Department of Biostatistics and Data Science, Wake Forest School of Medicine, Winston-Salem, NC, USA. [76]Medical Research Council Human Genetics Unit, Institute of Genetics and Cancer, University of Edinburgh, Edinburgh, UK. [77]Department of Genetics, University of North Carolina, Chapel Hill, NC, USA. [78]Department of Biostatistics and Epidemiology, University of Massachusetts Amherst, Amherst, MA, USA. [79]Vanderbilt University Medical Center, Division of Nephrology and Hypertension, Vanderbilt Center for Kidney Disease and Integrated Program for Acute Kidney Injury Research, and Vanderbilt Precision Nephrology Program Nashville, Nashville, TN, USA. [80]Nuffield Department of Medicine, University of Oxford, Oxford, UK. [81]Broad Institute of Harvard and MIT, Cambridge, MA, USA. [82]Big Data Institute, Li Ka Shing Centre for Health Information and Discovery, University of Oxford, Oxford OX3 7LF, UK. [83]Genome Institute of Singapore, Agency for Science Technology and Research, Singapore, Singapore. [84]School of Public Health, Tongji Medical College, Huazhong University of Science and Technology, Wuhan, China. [85]Singapore Eye Research Institute, Singapore National Eye Center, Singapore, Singapore. [86]Ophthalmology and Visual Sciences Academic Clinical Program (Eye ACP), Duke—NUS Medical School, Singapore, Singapore. [87]Department of Paediatrics, Yong Loo Lin School of Medicine, National University of Singapore, Singapore, Singapore. [88]Khoo Teck Puat–National University Children's Medical Institute, National University Health System, Singapore, Singapore. [89]Nephrology and Dialysis Unit, Genomics of Renal Diseases and Hypertension Unit, IRCCS San Raffaele Scientific Institute, Milan, Italy. [90]Department of Ophthalmology, Yong Loo Lin School of Medicine, National University of Singapore and National University Health System, Singapore, Singapore. [91]Research Unit Molecular Epidemiology, Helmholtz Zentrum München—German Research Center for Environmental Health, Neuherberg, Germany. [92]Department of Biomedical Informatics, Vanderbilt University Medical Center, Nashville, TN, USA. [93]Department of Clincial Sciences in Malmö, Lund University, Malmö, Sweden. [94]Department of Internal Medicine, Division of Cardiology, University of Michigan, Ann Arbor, MI 48109, USA. [95]Department of Computational Medicine and Bioinformatics, University of Michigan, Ann Arbor, MI 48109, USA. [96]Department of Human Genetics, University of Michigan, Ann Arbor, MI 48109, USA. [97]Division of Preventive Medicine, Brigham and Women's Hospital, Boston, MA, USA. [98]Harvard Medical School, Boston, MA, USA. [99]deCODE Genetics/Amgen, Inc., Reykjavik, Iceland. [100]Iceland School of Engineering and Natural Sciences, University of Iceland, Reykjavik, Iceland. [101]IRCCS Neuromed, Pozzilli, Italy. [102]San Raffaele Research Institute, Milan, Italy. [103]Department of Translational Research in Psychiatry, Max Planck Institute of Psychiatry, Munich, Germany. [104]Center for Cognitive Ageing and Cognitive Epidemiology, University of Edinburgh, Edinburgh, UK. [105]Centre for Research Ethics & Bioethics, Department of Public Health and Caring Sciences, Uppsala University, Uppsala, Sweden. [106]Department of Public Health and Primary Care, Leiden University Medical Center, Leiden, The Netherlands. [107]Department of Clinical Epidemiology, Leiden University Medical Center, Leiden, The Netherlands. [108]Division of Statistical Genomics, Department of Genetics, Washington University School of Medicine, St. Louis, MO, USA. [109]Clinical Division of Neurogeriatrics, Department of Neurology, Medical University of Graz, Graz, Austria. [110]Institute for Medical Informatics, Statistics and Documentation, Medical University of Graz, Graz, Austria. [111]Human Genetics Center, University of Texas Health Science Center, Houston, TX, USA. [112]Neuroalgology Unit, Fondazione IRCCS Istituto Neurologico 'Carlo Besta', Milan, Italy. [113]Charles Bronfman Institute for Personalized Medicine, Icahn School of Medicine at Mount Sinai, New York, NY, USA. [114]Digital Health Center, Hasso Plattner Institute and University of Potsdam, Potsdam, Germany. [115]Department of Medicine, Yong Loo Lin School of Medicine, National University of Singapore and National University Health System, Singapore, Singapore. [116]Duke - NUS Medical School, Singapore, Singapore. [117]ePhood Scientific Unit, ePhood SRL, Milano, Italy. [118]Department of Internal Medicine, Erasmus MC, University Medical Center Rotterdam, Rotterdam, The Netherlands. [119]Generation R Study Group, Erasmus MC, University Medical Center Rotterdam, Rotterdam, The Netherlands. [120]Cardiology, Geneva University Hospitals, Geneva, Switzerland. [121]Department of Medicine, Internal Medicine, Lausanne University Hospital and University of Lausanne, Lausanne, Switzerland. [122]Institute of Molecular Genetics "Luigi Luca Cavalli-Sforza", National Research Council of Italy, Pavia, Italy. [123]Department of Psychiatry, University Hospital of Lausanne, Lausanne, Switzerland. [124]Division of Nephrology, Department of Medicine, Icahn School of Medicine at Mount Sinai, New York, NY, USA. [125]Vth Department of Medicine (Nephrology, Hypertensiology, Rheumatology, Endocrinology, Diabetology), Medical Faculty Mannheim, University of Heidelberg, Mannheim, Germany. [126]Institute for Molecular Bioscience, University of Queensland, St Lucia, QLD, Australia. [127]Centre for Global Health, Usher Institute, University of Edinburgh, Edinburgh, UK. [128]Green Lane Cardiovascular Service, Auckland City Hospital and University of Auckland, Auckland, New Zealand. [129]Department of Biostatistics and Center for Statistical Genetics, University of Michigan, Ann Arbor, MI, USA. [130]Research Unit Genetic Epidemiology, Gottfried Schatz Research Center for Cell Signaling, Metabolism and Aging, Medical University of Graz, Graz, Austria. [131]Division of Nephrology and Hypertension, Loyola University Chicago, Chicago, IL, USA. [132]Department of Hygiene and Epidemiology, University of Ioannina Medical School, Ioannina, Greece. [133]Department of Clinical Biochemistry, Landspitali University Hospital, Reykjavik, Iceland. [134]Department of Nephrology and Medical Intensive Care, Charité—Universitätsmedizin Berlin, Berlin, Germany.

[135]Department of Pediatrics, Erasmus MC, University Medical Center Rotterdam, Rotterdam, The Netherlands. [136]Department of Cardiology, Ealing Hospital, London North West University Healthcare NHS Trust, Middlesex, UK. [137]Imperial College Healthcare NHS Trust, Imperial College London, London, UK. [138]MRC–PHE Center for Environment and Health, School of Public Health, Imperial College London, London, UK. [139]National Heart and Lung Institute, Imperial College London, London, UK. [140]The Institute for Translational Genomics and Population Sciences, Department of Pediatrics, The Lundquist Institutefor Biomedical Innovation at Harbor-UCLA Medical Center, Torrance, CA, USA. [141]Beijing Institute of Ophthalmology, Beijing Key Laboratory of Ophthalmology and Visual Sciences, Beijing Tongren Hospital, Capital Medical University, Beijing, China. [142]Institute of Laboratory Medicine, Clinical Chemistry and Molecular Diagnostics, University of Leipzig, Leipzig, Germany. [143]Division of Family Medicine and Primary Care, Department of Neurobiology, Care Sciences and Society, Karolinska Institutet, Stockholm, Sweden. [144]School of Health and Social Studies, Dalarna University, Stockholm, Sweden. [145]University of Eastern Finland, Kuopio, Finland. [146]Kuopio University Hospital, Kuopio, Finland. [147]Icelandic Heart Association, Kopavogur, Iceland. [148]The Center of Public Health Sciences, University of Iceland, Reykjavík, Iceland. [149]Montreal University Hospital Research Center, CHUM, Montreal, QC, Canada. [150]CRCHUM, Montreal, QC, Canada. [151]Lee Kong Chian School of Medicine, Nanyang Technological University, Singapore, Singapore. [152]QIMR Berghofer Medical Research Institute, Brisbane, QLD, Australia. [153]Department of Internal Medicine, Harvard Medical School, Boston, MA, USA. [154]VA Cooperative Studies Program, VA Boston Healthcare System, Boston, MA, USA. [155]Department of Ophthalmology, Medical Faculty Mannheim, University Heidelberg, Mannheim, Germany. [156]Instituteof Molecular and Clinical Ophthalmology, Basel, Switzerland. [157]Privatpraxis Prof Jonas und Dr Panda-Jonas, Heidelberg, Germany. [158]Center for Public Health Genomics, University of Virginia, Charlottesville, Charlottesville, VA, USA. [159]Danish Aging Research Center, University of Southern Denmark, Odense C, Denmark. [160]Intensive Care Medicine, Charité, Berlin, Germany. [161]Department of Nephrology and Hypertension, Friedrich Alexander University Erlangen-Nürnberg (FAU), Erlangen, Germany. [162]Department of Anatomy and Cell Biology, University Medicine Greifswald, Greifswald, Germany. [163]Division of Endocrinology, Diabetes and Nutrition, University of Maryland School of Medicine, Baltimore, MD, USA. [164]Department of Biostatistics, University of Washington, Seattle, WA, USA. [165]Geisinger Research, Biomedical and Translational Informatics Institute, Rockville, MD, USA. [166]Department of Nephrology, Geisinger, Danville, PA, USA. [167]Department of Cardiology, Heart Center, Tampere University Hospital, Tampere, Finland. [168]Department of Cardiology, Finnish Cardiovascular Research Center—Tampere, Faculty of Medicine and Health Technology, Tampere University, Tampere, Finland. [169]Laboratory of Clinical Genome Sequencing, Graduate School of Frontier Sciences, The University of Tokyo, Tokyo, Japan. [170]Institute of Genetic Epidemiology, Helmholtz Zentrum München—German Research Center for Environmental Health, Neuherberg, Germany. [171]Chair of Genetic Epidemiology, IBE, Faculty of Medicine, Ludwig-Maximilians-Universität München, München, Germany. [172]Institute of Medical Biostatistics, Epidemiology and Informatics (IMBEI), University Medical Center, Johannes Gutenberg University, Mainz, Germany. [173]Cardiovascular Epidemiology, Department of Medical Sciences, Uppsala University, Uppsala, Sweden. [174]Cardiology, Department of Medical Sciences, Uppsala University, Uppsala, Sweden. [175]Uppsala Clinical Research Center, Uppsala University, Uppsala, Sweden. [176]Atlanta VA Health Care System, Decatur, GA, USA. [177]Division of Endocrinology and Metabolism, Department of Medicine, Emory University School of Medicine, Atlanta, GA, USA. [178]Laboratory of Epidemiology and Population Sciences, National Institute on Aging, Intramural Research Program, US National Institutes of Health, Bethesda, MD, USA. [179]Department of Clinical Chemistry, Fimlab Laboratories, Tampere, Finland. [180]Department of Clinical Chemistry, Finnish Cardiovascular Research Center—Tampere, Faculty of Medicine and Health Technology, Tampere University, Tampere, Finland. [181]Division of Biomedical Informatics and Personalized Medicine, School of Medicine, University of Colorado Denver–Anschutz Medical Campus, Aurora, CO, USA. [182]Department of Epidemiology, Erasmus MC, University Medical Center Rotterdam, Rotterdam, The Netherlands. [183]Max Planck Institute of Psychiatry, Munich, Germany. [184]SYNLAB MVZ Humangenetik Mannheim, Mannheim, Germany. [185]Institute of Genetic and Biomedical Research, National Research Council of Italy, Cagliari, Italy. [186]Finnish Institute for Health and Welfare, Helsinki, Finland. [187]Research Program for Clinical and Molecular Metabolism, Faculty of Medicine, University of Helsinki, Helsinki, Finland. [188]Division of Nephrology, Department of Internal Medicine, University of Groningen, University Medical Center Groningen, Groningen, The Netherlands. [189]Target Sciences—Genetics, GlaxoSmithKline, Albuquerque, NM, USA. [190]Laboratory for Statistical Analysis, RIKEN Center for Integrative Medical Sciences (IMS), Yokohama, Japan. [191]Department of Biomedical Informatics, Harvard Medical School, Boston, MA, USA. [192]Department of Ophthalmology, Graduate School of Medical Sciences, Kyushu University, Fukuoka, Japan. [193]Department of Ophthalmology, Tohoku University Graduate School of Medicine, Miyagi, Japan. [194]Institute of Clinical Chemistry and Laboratory Medicine, University Medicine Greifswald, Greifswald, Germany. [195]DZHK (German Center for Cardiovascular Research), Partner Site Munich Heart Alliance, Munich, Germany. [196]Cardiovascular Division, Brigham and Women's Hospital, Boston, MA, USA. [197]TIMI Study Group, Boston, MA, USA. [198]RIKEN Center for Integrative Medical Sciences (IMS), Yokohama (Kanagawa), Japan. [199]Department of Clinical Physiology, Tampere University Hospital, Tampere, Finland. [200]Department of Clinical Physiology, Finnish Cardiovascular Research Center—Tampere, Faculty of Medicine and Health Technology, Tampere University, Tampere, Finland. [201]Laboratory of Neurogenetics, National Institute on Aging, National Institutes of Health, Bethesda, MD, USA. [202]Data Tecnica International, Glen Echo, MD, USA. [203]The Department of Public Health and Welfare, Finnish Institute for Health and Welfare, Helsinki, Finland. [204]South Texas Diabetes and Obesity Institute and Department of Human Genetics, University of Texas Rio Grande Valley School of Medicine, Brownsville, TX, USA. [205]Department of Genetics, School of Medicine, Mashhad University of Medical Sciences, Mashhad, Iran. [206]Center for Primary Care and Public Health (Unisanté), University of Lausanne, 1010 Lausanne, Switzerland. [207]Department of Population Health Sciences, Geisinger Health, 100 N. Academy Ave., Danville, PA, USA. [208]Biochemistry, Wake Forest School of Medicine, Winston-Salem, NC, USA. [209]Taub Institute for Research on Alzheimer's Disease and the Aging Brain, Columbia University Medical Center, New York, NY, USA. [210]Department of Cardiology, University of Groningen, University Medical Center Groningen, Groningen, The Netherlands. [211]Tampere Centre for Skills Training and Simulation, Faculty of Medicine and Health Technology, Tampere University, Tampere, Finland. [212]Division of Nephrology, University of Washington, Seattle, WA, USA. [213]Kidney Research Institute, University of Washington, Seattle, WA, USA. [214]Institute of Physiology, University of Zurich, Zurich, Switzerland. [215]Department of Clinical Physiology and Nuclear Medicine, Turku University Hospital, Turku, Finland. [216]Research Center of Applied and Preventive Cardiovascular Medicine, University of Turku, Turku, Finland. [217]Centre for Population Health Research, University of Turku and Turku University Hospital, Turku, Finland. [218]Faculty of Medicine, University of Split, Split, Croatia. [219]Algebra University College, Ilica 242, Zagreb, Croatia. [220]Department of Medical Epidemiology and Biostatistics, Karolinska Institutet, Stockholm, Sweden. [221]Imperial College NIHR Biomedical Research Center, Imperial College London, London, UK. [222]Health Data Research UK—London, London, UK. [223]Medpharmgene, Montreal, QC, Canada. [224]Department of Clinical Science and Education, Karolinska Institutet, Södersjukhuset, Stockholm, Sweden. [225]Department of Cardiology, Södersjukhuset, Stockholm, Sweden. [226]Integrated Research and Treatment Center Adiposity Diseases, University of Leipzig, Leipzig, Germany. [227]Steno Diabetes Center Copenhagen, Herlev, Denmark. [228]Department of Clinical Medicine, Faculty of Health and Medical Sciences, University of Copenhagen, Copenhagen, Denmark. [229]Department of Genetics, University of Groningen, University Medical Center Groningen, Groningen, The Netherlands. [230]Institute of Clinical Chemistry and Laboratory Medicine, University Hospital Regensburg, Regensburg, Germany. [231]Vanderbilt Genetics Institute, Vanderbilt University Medical Center, Nashville, TN, USA. [232]Section of Gerontology and Geriatrics, Department of Internal Medicine, Leiden University Medical Center, Leiden, The Netherlands. [233]Department of Computational Biology, University of Lausanne, Lausanne, Switzerland. [234]Swiss Institute of

Bioinformatics, Lausanne, Switzerland. [235]Department of Exercise and Nutrition Sciences, Milken Institute School of Public Health, The George Washington University, Washington, DC, USA. [236]The Mindich Child Health and Development Institute, Icahn School of Medicine at Mount Sinai, New York, NY, USA. [237]Novo Nordisk Foundation Center for Basic Metabolic Research, Department of Health and Medical Sciences, University of Copenhagen, Copenhagen, Denmark. [238]Institute of Cardiovascular and Medical Sciences, University of Glasgow, Glasgow, UK. [239]Institute of Medical Informatics and Statistics, Kiel University, University Hospital Schleswig-Holstein, Kiel, Germany. [240]Geisinger Research, Biomedical and Translational Informatics Institute, Danville, PA, USA. [241]NHLBI's Framingham Heart Study, Framingham, MA, USA. [242]The Center for Population Studies, NHLBI, Framingham, MA, USA. [243]Department of Nephrology, St. Olavs Hospital, Trondheim University Hospital, Trondheim, Norway. [244]Diabetes Center, Khoo Teck Puat Hospital, Singapore, Singapore. [245]The Bioinformatics Center, Department of Biology, Faculty of Science, University of Copenhagen, Copenhagen, Denmark. [246]Institute of Human Genetics, Helmholtz Zentrum München, Neuherberg, Germany. [247]Institute of Human Genetics, Technische Universität München, Munich, Germany. [248]Einthoven Laboratory of Experimental Vascular Research, Leiden University Medical Center, Leiden, The Netherlands. [249]Program in Medical and Population Genetics, Broad Institute, Cambridge, MA, USA. [250]Interfaculty Institute for Genetics and Functional Genomics, University Medicine Greifswald, Greifswald, Germany. [251]Molecular Geriatrics, Department of Public Health and Caring Sciences, Uppsala University, Uppsala, Sweden. [252]Department of Genetics, Shanghai—MOST Key Laboratory of Health and Disease Genomics, Chinese National Human Genome Center, Shanghai, China. [253]Shanghai Industrial Technology Institute, Shanghai, China. [254]Beijing Tongren Eye Center, Beijing Tongren Hospital, Capital Medical University, Beijing, China. [255]Synlab Academy, Synlab Holding Deutschland GmbH, Mannheim, Germany. [256]Clinical Institute of Medical and Chemical Laboratory Diagnostics, Medical University of Graz, Graz, Austria. [257]Deutsches Herzzentrum München, Technische Universität München, Munich, Germany. [258]Institute of Epidemiology and Medical Biometry, University of Ulm, Ulm, Germany. [259]Institute of Epidemiology and Biobank Popgen, Kiel University, Kiel, Germany. [260]School of Public Health and Community Medicine, Hebrew University of Jerusalem, Jerusalem, Israel. [261]Laboratory of Complex Trait Genomics, Department of Computational Biology and Medical Sciences, Graduate School of Frontier Sciences, The University of Tokyo, Tokyo, Japan. [262]Laboratory for Systems Genetics, RIKEN Center for Integrative Medical Sciences (IMS), Osaka, Japan. [263]Department of Statistical Genetics, Osaka University Graduate School of Medicine, Osaka, Japan. [264]Massachusetts General Hospital, Boston, MA, USA. [265]Department of Genetics and Pharmacology, Institute of Genetic Epidemiology, Medical University of Innsbruck, Innsbruck, Austria. [266]These authors contributed equally: Thomas W. Winkler, Humaira Rasheed, Alexander Teumer, Mathias Gorski, Bryce X. Rowan. [267]These authors jointly supervised this work: Adriana M. Hung, Florian Kronenberg, Anna Köttgen, Cristian Pattaro, Iris M. Heid. *Lists of authors and their affiliations appear at the end of the paper.
✉email: thomas.winkler@ukr.de; iris.heid@ukr.de

## Lifelines cohort study

Chris H. L. Thio[19], Peter J. van der Most[19] & Martin H. de Borst[189]

## DiscovEHR/MyCode study

Kevin Ho[166,167], Navya Shilpa Josyula[208] & Sarah A. Pendergrass[240]

## VA Million Veteran Program

Bryce X. Rowan[9,10], Cassianne Robinson-Cohen[10,80], John M. Gaziano[154,155], Lawrence S. Phillips[177,178], Ran Tao[9,232] & Adriana M. Hung[10,80]

A full List of members and their affiliations appears in the Supplementary Information.

