## [Peer Review File · Communications Biology]

Reviewers' comments:

Reviewer #1 (Remarks to the Author):

The paper by Winkler et al describes a comprehensive approach to dissect DM and no-DM genetics effects on eGFR. The methods are sound, and the combination of different approaches increases the robustness of the findings. The 32 novel DM-specific loci and the 11 loci with significant or suggestive differential DM/noDM effect shed light on the differences in the genetic architecture of eGFR in the two populations. Finally, the gene-prioritization study identifies some potential druggable targets, while the GRS study shows no advantage of including DM/onDM specific effects for genetic prediction in the two patient populations (DM vs no DM).

While I find it is sometimes difficult to follow the results section, the flowchart (Figure 1) is helpful- the authors should make sure to redirect the reader to the flowchart when needed.

My comments are minor:

I was wondering if the authors looked at sub-populations of DM patients (type 1 vs type 2 diabetes), given the distinct genetic architecture of the two traits. Are their differential genetic effects on the nephropathy due to type 1 vs type 2 diabetes? The authors looked at genetic overlap of the eGFR loci with type 2 diabetes, but it would be worth doing the same for type 1 diabetes.

Results, GRS paragraph, line 570: "This may be attributable to the larger uncertainty in effect size estimation in DM due to the smaller sample size". I am not sure I understand this. Was not the sample size (ie DM sample) identical to the size testing the effects of the GRS with non-specific weights

Discussion: is there a plausible pathophysiological explanation of effects of cubilin on kidney function? The same for RASSF6 and PGAP3.

Discussion: The authors should mention that the results of this European GWAS should be replicated in other ancestries. Given the differences in the prevalence of DM among Europeans and non-Europeans, with certain ethnic group being more at risk of DM, such studies could yield interesting results.

Methods, line 591: "Our gene prioritization at identified difference or novel loci" : this phrase does not read well, please rephrase.

Line 676: please correct typo ("we their used")

Line 684: Definition of diabetes : to establish a diagnosis of diabetes, a single abnormal glucose or Hba1c measurement is not enough in the absence of symptomatic hyperglycemia. See definition of DM in Table 3 in the link below:

<https://guidelines.diabetes.ca/cpg/chapter3>

"In the absence of symptomatic hyperglycemia, if a single laboratory test result is in the diabetes range, a repeat confirmatory laboratory test (FPG, A1C, 2hPG in a 75 g OGTT) must be done on another day. It is preferable that the same test be repeated (in a timely fashion) for confirmation". Thus, the definition used here might have slightly overestimated the DM cases.

Reviewer #2 (Remarks to the Author):

In this paper, the authors conducted a large-scale genetic association analysis to investigate genetic variants differentially associated with kidney function in diabetic or non-diabetic individuals. A solid study design with a powerful replication dataset was applied. I believe the external validity of the results should be high, and some of them showed strong associations. My overall impression was that the description was technically precise. Still, it would have been helpful if more analysis or description of the clinical/functional implications of the discovered loci/variants so that the reader could better understand the importance and significance of these discoveries. The followings are my comments.

[Major comments]

1. Additional analysis regarding diabetes and surrogate outcomes regarding diabetic nephropathy leveraging large-scale biobanks included in this study might be helpful for further characterization

of these associated variants.

First, a more explicit description regarding diabetes would be helpful to understand pathophysiology. In particular, the concordance or discordance of the directions of the effects between DM-specific eGFR effects and DM prevalence is vital to understanding pathophysiology. The current description seems to contain some ambiguity (For example, "near MED1/NEUROD2 and UMOD/PDILT, eGFR-lowering allele as DM-risk decreasing").

Also, in this study, the discovery analysis was made by stratifying the population by the binary outcome - presence or absence of diabetes. It will be interesting to see if similar associations and dose-response can be observed when diabetes is considered a continuous value using HbA1c leveraging large biobanks included in this study as a complementary analysis.

2. A phewas-like approach for these eGFR associated DM/non-DM differential variants and its interpretation might make the manuscript more informative. A quick search shows that rs77924615 or rs9928003 at the UMOD-PDILT locus have been reported to be associated with a variety of traits.

https://genetics.opentargets.org/variant/16_20381010_G_A

https://genetics.opentargets.org/variant/16_20346926_A_C

Are there any clues to understanding the mechanism of worsening renal function specific to diabetic/non-diabetic by evaluating these pleiotropic effects of the variants the authors have discovered?

3. Regarding GRS analyses, I wondered if the inclusion of many variants related to eGFR (>600) might have blunted the associations. Is it possible to evaluate that the GRS focusing only on the differentially associated variants detected in this study explicitly contributes to the improvement of prediction accuracy of renal function in diabetes?

[Minor comments]

The manuscript would be clearer if the authors included the information about the lead variant (rsID, minor allele frequency, beta, se) and the name of the locus in the text.

Please describe the statistics of the GWAS inflation; is the lambda GC of the combined meta-analysis for eGFR reasonable in comparison with previous studies?

How did you deal with variants showing extremely high beta or heterogeneity in the eGFR meta-analysis?

What is the rationale for assuming "the two independent index variants rs77924615 or rs34882080" in Supplementary Fig 2?

Regarding line 421, "despite the multiple independent variants known for eGFR association overall (Supplementary Figure 4A, B)", please cite a reference or provide data showing multiple signals detected in eGFR.

Reviewer #1

The paper by Winkler et al describes a comprehensive approach to dissect DM and no-DM genetics effects on eGFR. The methods are sound, and the combination of different approaches increases the robustness of the findings. The 32 novel DM-specific loci and the 11 loci with significant or suggestive differential DM/noDM effect shed light on the differences in the genetic architecture of eGFR in the two populations. Finally, the gene-prioritization study identifies some potential druggable targets, while the GRS study shows no advantage of including DM/onDM specific effects for genetic prediction in the two patient populations (DM vs no DM).

Response: We thank the reviewer for the very clear and constructive comments. We have addressed the comments below point-by-point and revised the manuscript accordingly.

While I find it is sometimes difficult to follow the results section, the flowchart (Figure 1) is helpful- the authors should make sure to redirect the reader to the flowchart when needed.

Response: We agree with the reviewer that the analysis flow was difficult to follow at some points. We went through the whole results section and tried to improve general readability, e.g. by careful re-wording of difference and joint test results (page 9-12). In addition, as suggested, we have added further references to Figure 1 at the beginning of respective sections, i.e. page 9 (first and last paragraph (par.)), page 11 (2nd). Additionally, we have structured Figure 1 into Figure 1A-D to help the reader navigate the Results and Figure.

My comments are minor:

I was wondering if the authors looked at sub-populations of DM patients (type 1 vs type 2 diabetes), given the distinct genetic architecture of the two traits. Are their differential genetic effects on the nephropathy due to type 1 vs type 2 diabetes? The authors looked at genetic overlap of the eGFR loci with type 2 diabetes, but it would be worth doing the same for type 1 diabetes.

Response: We agree that a better distinction of Type 1 and Type 2 Diabetes would be helpful. However, we cannot make this distinction in the here-collected study-specific summary statistics. Due to the fact that most of the studies included in the meta-analysis are population-based with study-specific mean age >40 years, the studies reflect Type 2 and Type 1 Diabetes according to population prevalence. Thus, ~90-95% of the DM-individuals here are Type 2. We have expanded the discussion (page 17):

These DM-individuals were mostly from population-based studies of adult individuals. As such, the proportion of type 1 DM among DM-individuals analyzed here reflects the proportion of type 1 DM of 5-10% among adult DM-individuals⁵⁹. Due to the substantially larger number of type 2 DM individuals, the identified DM/noDM-differential effects on eGFR may mostly reflect differences between type 2 DM versus noDM. A better distinction by type of DM would require more granular data focusing on the distinction between these two major diabetes groups and substantially larger data on type 1 DM.

As suggested, we now queried the association of our 11 DM/noDM-difference lead variants with type 1 DM from a recently published type 1 diabetes GWAS (Chiou et al. *Nature* 2021). None of the 11

variants showed a significant association with type 1 DM. We approached this aspect of overlap with reported type 1 DM association in 2 ways: we explored the overlap between identified DM/noDM difference signals with reported loci (genome-wide significant, $P < 5 \times 10^{-8}$) for type 1 DM and we reported the 11-variants' association with type 1 DM judged at Bonferroni-corrected significance for the 11 variants.

We added this finding to the results (page 12) and to **Supplementary Table 10**:

“We were also interested whether any of the 11 loci with DM/noDM-difference overlapped with known genome-wide significant loci for type 1 DM⁴², type 2 DM¹⁹ or glyceamic traits⁴³. None of the 11 loci overlapped with type 1 DM, type 2 DM, or glyceamic traits loci (all 11 lead variants $P > 5.0 \times 10^{-8}$ in published GWAS for DM-risk, glucose or insulin levels, **Supplementary Table 10**). None of the 11 variants was associated with type 1 DM judged at Bonferroni-corrected significance ($P > 0.05/11 = 4.5 \times 10^{-3}$). Three of the 11 variants were associated with type 2 DM ($P < 0.05/11 = 4.5 \times 10^{-3}$, rs77924615 near *UMOD-PDILT*, rs55722796 near *MED1-NEUROD2*) or fasting glucose (rs963837 near *DCDC5*).”

Results, GRS paragraph, line 570: “This may be attributable to the larger uncertainty in effect size estimation in DM due to the smaller sample size”. I am not sure I understand this. Was not the sample size (ie DM sample) identical to the size testing the effects of the GRS with non-specific weights

Response: We apologize for the confusion and agree that the statement was misleading. Indeed, the sample size for testing the various versions of the GRS was identical within HUNT, e.g., for GRS in DM, always $n=3,023$, i.e. the DM sample size in the HUNT study. However, the sample size to estimate DM-specific effects versus noDM-specific effects, which are used as weights in the weighted GRS, is substantially smaller for DM. We have expanded these analyses now also to a 7-variant risk score, focused on DM-/noDM-differential effect lead variants according to another reviewer's suggestion. To accommodate this and to improve clarity with regard to this Reviewer's question, we have re-worded the section to make this clearer (page 15):

When comparing different weighting schemes, we found no notable improvement in R^2 values by using DM-/noDM-specific weights compared to overall weights (**Table 3**). In the 634-variant GRS, using DM-specific weights even reduced R^2 in DM individuals ($R^2=3.1\%$ versus 4.0%). This may be attributable to the larger uncertainty in DM-specific weights estimated from the meta-analysis restricted to DM individuals.

Discussion: is there a plausible pathophysiological explanation of effects of cubilin on kidney function? The same for *RASSF6* and *PGAP3*.

Response: We agree that these are interesting genes and have extended the discussion on these substantially (page 16):

“Particularly interesting was *CUBN* in a novel eGFR locus. The *CUBN* locus is the GWAS locus with the strongest effect on UACR⁵ and microalbuminuria⁶ and identified here for eGFR at genome-wide significance, but respective lead variants for eGFR and UACR are uncorrelated. The lead variant for

eGFR, rs11254238, is highly correlated with a variant that alters the encoded protein, cubilin (rs1801232, $r^2=0.91$). We observed a twice as large effect on eGFR in DM compared to noDM in both discovery and replication data, but not statistically significantly different. This is in line with previously observed larger *CUBN* variants' effects on microalbuminuria⁶ and UACR^{7,8} in DM, which provides further evidence for an interaction of *CUBN* with DM-status on kidney function. In fact, our previous sequencing of the *CUBN* gene found rare variants in *CUBN* that were associated with higher UACR and with better eGFR⁹. Our study now reports such a parallel association on UACR and eGFR now also for a common variant at the genome-wide significance levels. Together, our results provide further support of the importance to assess the physiological role of *CUBN* by functional studies not only for microalbuminuria, but also for finding pathophysiological explanations related to impaired filtration rate.

The *RASSF6* gene has been studied in relation to the kidney as well: *RASSF6* mediates apoptosis in various cells¹⁰. It is shown to be expressed in the slit diaphragm in glomeruli and the apical membranes in proximal renal tubular epithelial cells of rat kidney. The same animal study suggests involvement of the *RASSF6* pathway in contrast-induced nephropathy. Thus, the novel detection of the *RASSF6* locus for eGFR here and the lead variant, rs17804499, being protein-altering and with 86% probability causal render *RASSF6* a compelling candidate for functional follow-up to help understand its role in kidney function that is elusive so far.

Another interesting gene was *PGAP3* in the *MED1-NEUROD2* locus, which mapped to an expression-modulating variant that was associated with eGFR in noDM, but not in DM individuals. *PGAP3* is a known gene for CKD and eGFR: eQTL analyses showed colocalization of the association signal with *PGAP3* expression¹¹. *PGAP3* knockout mice developed larger glomeruli with deposition of immunoglobulins¹², although we acknowledge that this does not provide a mechanistic explanation for its association with eGFR. Eventually, if causal genes at eGFR loci with significant differences by DM status will be advanced as drug targets for reno-protective therapies, the knowledge about DM- or noDM-specificity of the eGFR association can help define the target group for a potential subsequent therapy, as DM-only, noDM-only, or both."

Discussion: The authors should mention that the results of this European GWAS should be replicated in other ancestries. Given the differences in the prevalence of DM among Europeans and non-Europeans, with certain ethnic group being more at risk of DM, such studies could yield interesting results.

Response: We fully agree that additional analyses in non-EUR cohorts will be essential and important. We have added this to the discussion (page 18):

This GWAS included mostly European individuals and our findings require replication in non-European ancestries⁶³, particularly because of the large differences in DM prevalence across ancestries⁶⁴.

Methods, line 591: "Our gene prioritization at identified difference or novel loci" : this phrase does not read well, please rephrase.

Response: We agree and have reworded as follows:

Our gene prioritization at the 11 loci with DM/noDM-difference and at the 28 other novel eGFR loci highlighted 18 genes:

Line 676: please correct typo (“we their used”)

Response: Thank you. This was now corrected.

Line 684: Definition of diabetes : to establish a diagnosis of diabetes, a single abnormal glucose or Hba1c measurement is not enough in the absence of symptomatic hyperglycemia. See definition of DM in Table 3 in the link below:

<https://guidelines.diabetes.ca/cpg/chapter3>

“In the absence of symptomatic hyperglycemia, if a single laboratory test result is in the diabetes range, a repeat confirmatory laboratory test (FPG, A1C, 2hPG in a 75 g OGTT) must be done on another day. It is preferable that the same test be repeated (in a timely fashion) for confirmation” .Thus, the definition used here might have slightly overestimated the DM cases.

Response: We thank the reviewer for bringing this up. We agree that a one-time Hba1C measurement is not a clinical diagnosis. Hba1C levels > 6.5% were used together with self-reported diabetes in UK Biobank according to a recent GWAS of diabetes (e.g., Mahajan et al 2018). We also agree that the DM definition used in the various studies implies a certain heterogeneity and may include some DM-individuals that are not clinically manifest DM – which is typical for most population-based studies. To address this, we have added this to the discussion (page 17):

While the use of self-reports or one-time measurements of glucose or HbA1c to define DM here is typical also for GWAS on DM^{19,60}, this implies heterogeneity in the DM definition and may include some individuals without clinically manifest DM.

Reviewer #2

In this paper, the authors conducted a large-scale genetic association analysis to investigate genetic variants differentially associated with kidney function in diabetic or non-diabetic individuals. A solid study design with a powerful replication dataset was applied. I believe the external validity of the results should be high, and some of them showed strong associations. My overall impression was that the description was technically precise. Still, it would have been helpful if more analysis or description of the clinical/functional implications of the discovered loci/variants so that the reader could better understand the importance and significance of these discoveries. The followings are my comments.

Response: We thank the reviewer for the overall favorable feedback and the very constructive comments. The comments gave us the opportunity to improve the manuscript. We have expanded analyses (please see below) and also the description of the clinical/function implications of the discovered loci/variants. Besides the specific points addressed in detail below, we have expanded the discussion particularly on 3 genes that we believe our data provides compelling clinical/function implications (discussion, page 17):

“Particularly interesting was *CUBN* in a novel eGFR locus. The *CUBN* locus is the GWAS locus with the strongest effect on UACR⁵ and microalbuminuria⁶ and identified here for eGFR at genome-wide significance, but respective lead variants for eGFR and UACR are uncorrelated. The lead variant for

eGFR, rs11254238, is highly correlated with a variant that alters the encoded protein, cubilin (rs1801232, $r^2=0.91$). We observed a twice as large effect on eGFR in DM compared to noDM in both discovery and replication data, but not statistically significantly different. This is in line with previously observed larger *CUBN* variants' effects on microalbuminuria⁶ and UACR^{7,8} in DM, which provides further evidence for an interaction of *CUBN* with DM-status on kidney function. In fact, our previous sequencing of the *CUBN* gene found rare variants in *CUBN* that were associated with higher UACR and with better eGFR⁹. Our study now reports such a parallel association on UACR and eGFR now also for a common variant at the genome-wide significance levels. Together, our results provide further support of the importance to assess the physiological role of *CUBN* by functional studies not only for microalbuminuria, but also for finding pathophysiological explanations related to impaired filtration rate.

The *RASSF6* gene has been studied in relation to the kidney as well: *RASSF6* mediates apoptosis in various cells¹⁰. It is shown to be expressed in the slit diaphragm in glomeruli and the apical membranes in proximal renal tubular epithelial cells of rat kidney. The same animal study suggests involvement of the *RASSF6* pathway in contrast-induced nephropathy. Thus, the novel detection of the *RASSF6* locus for eGFR here and the lead variant, rs17804499, being protein-altering and with 86% probability causal render *RASSF6* a compelling candidate for functional follow-up to help understand its role in kidney function that is elusive so far.

Another interesting gene was *PGAP3* in the *MED1-NEUROD2* locus, which mapped to an expression-modulating variant that was associated with eGFR in noDM, but not in DM individuals. *PGAP3* is a known gene for CKD and eGFR: eQTL analyses showed colocalization of the association signal with *PGAP3* expression¹¹. *PGAP3* knockout mice developed larger glomeruli with deposition of immunoglobulins¹², although we acknowledge that this does not provide a mechanistic explanation for its association with eGFR. Eventually, if causal genes at eGFR loci with significant differences by DM status will be advanced as drug targets for reno-protective therapies, the knowledge about DM- or noDM-specificity of the eGFR association can help define the target group for a potential subsequent therapy, as DM-only, noDM-only, or both."

[Major comments]

1. Additional analysis regarding diabetes and surrogate outcomes regarding diabetic nephropathy leveraging large-scale biobanks included in this study might be helpful for further characterization of these associated variants.

First, a more explicit description regarding diabetes would be helpful to understand pathophysiology. In particular, the concordance or discordance of the directions of the effects between DM-specific eGFR effects and DM prevalence is vital to understanding pathophysiology. The current description seems to contain some ambiguity (For example, "near *MED1/NEUROD2* and *UMOD/PDILT*, eGFR-lowering allele as DM-risk decreasing").

Response: We thank the reviewer for the constructive comment. We agree that the description of effect direction was confusing at some parts of the manuscript. We have revised the results section on DM-associations as follows (please note that the whole chapter was extended with regard to type 1 diabetes loci based on comments by reviewer 1, page 12):

"We were also interested whether any of the 11 loci with DM/noDM-difference overlapped with known genome-wide significant loci for type 1 DM⁴², type 2 DM¹⁹ or glycemic traits⁴³. None of the 11 loci overlapped with type 1 DM, type 2 DM, or glycemic traits loci (all 11 lead variants $P > 5.0 \times 10^{-8}$ in published GWAS for DM-risk, glucose or insulin levels, **Supplementary Table 10**). None of the 11

variants was associated with type 1 DM judged at Bonferroni-corrected significance ($P > 0.05/11 = 4.5 \times 10^{-3}$). Three of the 11 variants were associated with type 2 DM ($P < 0.05/11 = 4.5 \times 10^{-3}$, rs77924615 near *UMOD-PDILT*, rs55722796 near *MED1-NEUROD2*) or fasting glucose (rs963837 near *DCDC5*). For all three variants, the eGFR-lowering allele was associated with decreased type 2 DM risk or fasting-glucose. Thus, the SNP-effect on eGFR cannot be fully explained by DM-status, which would have yielded a DM-risk increasing or glucose-increasing effect by the eGFR-lowering allele. The observation is in line with a pleiotropic effect on DM and eGFR, but with adverse effect for one of the two (eGFR or DM/glucose) and a beneficial effect for the other, which should be considered in drug design when applicable. When taking-into-account the DM/no-DM-stratum where the eGFR-effect was more pronounced, the three variants consisted of (i) two variants with stronger eGFR-effects in DM (still significant eGFR effect in noDM, near *UMOD/PDILT* and *DCDC5*) and (ii) one variant with a noDM-only effect on eGFR (near *MED1-NEUROD2*)."

Also, in this study, the discovery analysis was made by stratifying the population by the binary outcome - presence or absence of diabetes. It will be interesting to see if similar associations and dose-response can be observed when diabetes is considered a continuous value using HbA1c leveraging large biobanks included in this study as a complementary analysis.

Response: We thank the reviewer for the interesting suggestion. We now include additional analyses in unrelated individuals of UK Biobank ($n = 368,005$), where we investigate interaction of the 11 difference variants with continuous HbA1c (centered at 5.35%; i.e., median of HbA1c in noDM individuals of UKB). For all 11 variants, the SNP main effects and SNP-by-HbA1c interaction effect sizes were directionally consistent with main and interaction effects sizes in the SNP-by-DM interaction analysis. This underscored again the negative interaction effect for the 10 of the 11 variants with larger effects in DM or DM-only effects, while the one variant (near *MED1/NEUROD2*) showed negative interaction effect for the SNP-by-DM as well as the SNP-by-HbA1c interaction. We have added a new **Supplementary Table 6** on the results and included the following text in the main Results section (page 12):

We explored the lead variants of the 11 eGFR loci identified with DM/noDM-difference (i.e. established/suggestive SNP-by-DM interaction) for SNP-by-HbA1c interaction in UK Biobank ($n=368,005$), making full use of the continuous variable HbA1c instead of binary DM-status. For all 11 variants, the SNP main effects and SNP-by-HbA1c interaction effect sizes were directionally consistent with main and interaction effects sizes in the SNP-by-DM interaction analysis (**Supplementary Table 6**). This underscored again the negative interaction effect for the 10 of the 11 variants with larger effects in DM or DM-only effects, while the one variant (near *MED1/NEUROD2*) showed negative interaction effect for the SNP-by-DM as well as the SNP-by-HbA1c interaction.

2. A phewas-like approach for these eGFR associated DM/non-DM differential variants and its interpretation might make the manuscript more informative. A quick search shows that rs77924615 or rs9928003 at the *UMOD-PDILT* locus have been reported to be associated with a variety of traits.

https://genetics.opentargets.org/variant/16_20381010 G A

https://genetics.opentargets.org/variant/16_20346926 A C

Are there any clues to understanding the mechanism of worsening renal function specific to diabetic/non-diabetic by evaluating these pleiotropic effects of the variants the authors have discovered?

Response: We agree this would be interesting. We now include a PheWAS-like lookup of other-trait associations based on the Open Targets Genetics data base for the 11 difference variants. The results are shown in the new **Supplementary Table 11** and in the Results section (page 13):

We also queried the Open Targets Genetics database¹⁶ for associations of the 11 variants with other traits. We found 126 genome-wide significant associations ($P < 5 \times 10^{-8}$, **Supplementary Table 11**), particularly for hypertension and blood counts: for 3 of the 11 variants, the eGFR-decreasing alleles were associated with increased risk of hypertension (near *UMOD/PDILT*, *DCDC5* and *PIK3CG*). The variant rs55722796 near *MED1/NEUROD2*, solely associated with eGFR in noDM-individuals, was not associated with hypertension, but the eGFR-decreasing allele (in noDM) was associated with decreased blood counts (i.e. decreased red blood cells, hemoglobin and haematocrit).

3. Regarding GRS analyses, I wondered if the inclusion of many variants related to eGFR (>600) might have blunted the associations. Is it possible to evaluate that the GRS focusing only on the differentially associated variants detected in this study explicitly contributes to the improvement of prediction accuracy of renal function in diabetes?

Response: This is an interesting aspect. We have explored this in the HUNT study. We used the 7 variants detected with significant DM-/noDM-difference and computed the same GRS analyses as for the 634 variants. We found that the GRS based on the 7 difference variants explained more of the eGFR variance in DM compared to noDM. This is consistent with the fact that 6 of the 7 difference variants displayed larger eGFR effects in DM compared to noDM. We have integrated the results in **Table 3** (page 34) and in the results section (page 15):

“Across all weighting schemes, we found no significant DM/noDM-difference in the 634-variant GRS association with eGFR ($P_{\text{Diff}} > 0.05$, **Table 3**). The 634-variant GRS explained more of the eGFR variance in noDM compared to DM (e.g., $R^2 = 6.0\%$ vs. 4.0% using overall-effect weights). Since the absolute 634-variant GRS effect was similar in DM and noDM (e.g., beta per $sd_{\text{GRS}} = -2.54$ versus -2.84 ml/min/1.73m² using overall-effect weights), this larger relative GRS effect in noDM can be attributed – at least in part - to the smaller eGFR variance in noDM versus DM (HUNT standard deviation = 11.8 versus 13.4 ml/min/1.73m², respectively). Yet, the 7-variant GRS explained more of the eGFR variance in DM compared to noDM (e.g., $R^2 = 0.98\%$ vs. 0.62% using overall-effect weights). The reason for this is that 6 of the 7 variants had larger eGFR-effects in DM than in noDM, which accumulated to a larger absolute GRS effect in DM (e.g., beta per $sd_{\text{GRS}} = -1.02$ in DM compared to -0.68 in noDM using overall-effect weights).

When comparing different weighting schemes, we found no notable improvement in R^2 values by using DM-/noDM-specific weights compared to overall weights (**Table 3**). In the 634-variant GRS, using DM-specific weights even reduced R^2 in DM individuals ($R^2 = 3.1\%$ versus 4.0%). This may be attributable to the larger uncertainty in DM-specific weights estimated from the meta-analysis restricted to DM individuals. For the 7-variant GRS, DM-specific weights slightly increased R^2 in DM individuals, but not markedly ($R^2 = 1.0\%$ versus 0.98%).”

[Minor comments]

The manuscript would be clearer if the authors included the information about the lead variant (rsID, minor allele frequency, beta, se) and the name of the locus in the text.

Response: We agree that this should be made clear. However, when we re-state the rs numbers at each mentioning of the locus lead variants, this is difficult to follow also. We have tried to make a compromise and added rs numbers, where the specific variants is of particular interest. Locus names are added throughout the text, where applicable. Please see e.g. results page 10 (2nd par.) or page 11 (last par.):

Please describe the statistics of the GWAS inflation; is the lambda GC of the combined meta-analysis for eGFR reasonable in comparison with previous studies?

Response: We agree that the GC lambda is an important metrics. The lambda GC for DM and noDM in the combined analyses were 1.03 and 1.15. These are reasonable in comparison with a GC lambda of 1.26, which was observed previously in an eGFR overall meta-analysis by Wuttke and colleagues (Nat Genet 2019). We performed a double GC correction and clarified this now in the Methods (page 20):

“In stage 1 of our analysis, separately in DM ($n_{DM}=88,829$) and noDM ($n_{noDM}=620,665$) strata, we conducted fixed-effect inverse-variance weighted meta-analyses of 72 GWAS of log(eGFR) using metal⁶⁸, and then meta-analyzed these results with DM/noDM-stratified GWAS from UKB ($n_{DM}=21,040$; $n_{noDM}=414,628$; European only). To adjust for population stratification within studies, we applied genomic control (GC) correction⁶⁹ to each study prior to the meta-analysis. We applied a second GC correction to the DM- and noDM stage 1 meta-analysis results (GC lambda = 1.02 and 1.20, respectively). We excluded variants that were present only in ≤ 36 stage 1 studies ($\leq 50\%$) and variants with cumulative minor allele count of < 400 in the stage 1 meta-analyses. In summary, 109,869 individuals with DM and 1,035,190 with noDM were included in stage 1. We followed variants identified at stage 1 in independent stage 2 meta-analyses. For stage 2, we included DM/noDM-stratified GWAS on log(eGFR) from MVP ($n_{DM}=57,430$, $n_{noDM}=122,966$, hospital-based), MGI ($n_{DM}=7,469$, $n_{noDM}=36,558$, hospital-based) and HUNT ($n_{DM}=3,799$, $n_{noDM}=65,590$, population-based), totalling 68,698 individuals with DM and 225,114 with noDM, all of European ancestry. Again, we applied study-specific GC-correction prior to the meta-analysis and a second GC correction⁶⁹ to the stage 2 meta-analysis results (GC lambdas = 1.00 and 1.02 for DM- and noDM, respectively). To maximize power for locus identification, we combined the double GC-corrected stage 1 and 2 meta-analysis separately by DM status via fixed-effect inverse-variance weighted meta-analyses of the two sources using metal⁶⁸. The GC lambda in this final meta-analysis was comparable to previous GWAS^{16,70} (GC lambdas = 1.03 and 1.15, respectively).”

How did you deal with variants showing extremely high beta or heterogeneity in the eGFR meta-analysis?

Response: Most large betas were excluded in the quality control step filtering on minor allele frequency or minor allele count (excluded variants with $MAF < 0.1\%$ per study and $MAC < 400$ in stage 1 meta-analysis, see methods page 21). As done previously (Wuttke et al., Nat Genet 2019), we did not further exclude variants based on large beta-estimates but checked identified variants for abnormal statistics. None of the identified lead variants displayed an abnormally large beta-estimate. We now include the heterogeneity I^2 and a heterogeneity P value (based on the CKDGen meta-analysis that included data from 72 studies) for the highlighted difference variants or the novel eGFR loci

joint/stratified test variants in the supplement (**Supplementary Table 2** for the difference variants; **Supplementary Table 7** for the novel eGFR loci). All variants display no or low between-study heterogeneity ($I^2 < 30\%$) except for the *UMOD/PDILT* variant. The *UMOD/PDILT* lead variant rs77924615 displays substantial between-study heterogeneity in the noDM meta-analysis ($I^2 = 48\%$, $p_{het} = 4.7e-8$; for comparison, in DM: $I^2 = 20.1\%$, $p_{het} = 0.08$). The I^2 in noDM is in line with the recent overall (non-stratified) eGFR meta-analysis by Wuttke et al. 2019, where this variant displayed an $I^2 > 60\%$. We state this in the manuscript in Methods (page 21):

For the identified variants, we assessed between-study heterogeneity based on the CKDGen meta-analysis using a Chi-Squared test and an I^2 statistic⁷² and verified the association statistics with regards to abnormal or unusually large effect sizes.

What is the rationale for assuming "the two independent index variants rs77924615 or rs34882080" in Supplementary Fig 2?

Response: We agree that this was not made fully clear. These two variants showed independent eGFR signals at the *UMOD/PDILT* locus demonstrated by conditional analyses in the previous eGFR meta-analysis by CKDGen (Wuttke et al Nat Genet 2019). We have added the reference to the figure legend (**Supplementary Figure 2**):

"We computed difference P-Values at the *UMOD/PDILT* locus unconditioned and conditioned for the two previously identified independent index variants rs77924615 or rs34882080², respectively, in stage 1 ($n_{DM}=109,993$, $n_{noDM}=1,070,999$)."

Regarding line 421, "despite the multiple independent variants known for eGFR association overall (Supplementary Figure 4A, B)", please cite a reference or provide data showing multiple signals detected in eGFR.

Response: We have added the references Wuttke et al Nat Genet 2019 and Stanzick et al Nat Commun 2021, which demonstrated 2 and 4 independent signals at the *UMOD/PDILT* locus, respectively (see results, page 10):

"Interestingly, there was also no further independent variant with DM/noDM-difference in the *UMOD-PDILT* locus besides rs77924715, despite the multiple independent variants known for eGFR association overall^{16,20} (**Supplementary Figure 4A,B**)."

REVIEWERS' COMMENTS:

Reviewer #1 (Remarks to the Author):

I thank the authors for addressing in detail my comments. I have no further suggestions.

Reviewer #2 (Remarks to the Author):

The revisions the authors have made to the manuscript are very effective and sufficient in addressing my concerns.

I appreciate your detailed revision and rebuttal comments.